# Integrated chemical and genetic screens unveil FSP1 mechanisms of ferroptosis regulation

Toshitaka Nakamura [1], Eikan Mishima [1,2], Naoya Yamada[1], André Santos Dias Mourão[3], Dietrich Trümbach [1], Sebastian Doll[1], Jonas Wanninger[1], Elena Lytton[1], Peter Sennhenn[4], Thamara Nishida Xavier da Silva [5], José Pedro Friedmann Angeli [5], Michael Sattler [3,6], Bettina Proneth [1,7] & Marcus Conrad [1,7] ✉

Ferroptosis, marked by iron-dependent lipid peroxidation, may present an Achilles heel for the treatment of cancers. Ferroptosis suppressor protein-1 (FSP1), as the second ferroptosis mainstay, efficiently prevents lipid peroxidation via NAD(P)H-dependent reduction of quinones. Because its molecular mechanisms have remained obscure, we studied numerous FSP1 mutations present in cancer or identified by untargeted random mutagenesis. This mutational analysis elucidates the FAD/NAD(P)H-binding site and proton-transfer function of FSP1, which emerged to be evolutionarily conserved among different NADH quinone reductases. Using random mutagenesis screens, we uncover the mechanism of action of next-generation FSP1 inhibitors. Our studies identify the binding pocket of the first FSP1 inhibitor, iFSP1, and introduce the first species-independent FSP1 inhibitor, targeting the NAD(P)H-binding pocket. Conclusively, our study provides new insights into the molecular functions of FSP1 and enables the rational design of FSP1 inhibitors targeting cancer cells.

Since the recognition of ferroptosis as a distinct iron-dependent form of cell death characterized by the oxidative destruction of cellular membranes, the process has attracted tremendous interest, likely owing to its high relevance in human diseases, such as neurodegenerative disorders, tissue ischemia–reperfusion injury and malignancies[1]. In particular, certain cancer-cell states, including cancer stem cells and therapy-resistant and disseminating cancer cells, have been reported to exhibit an inherent vulnerability to ferroptosis, providing a rationale for selective induction of ferroptosis as a next-generation cancer therapy approach[2–7]. Recently, FSP1 was identified as a powerful backup

system for a key regulator of ferroptosis called glutathione peroxidase 4 (GPX4)[8–11]. By reducing quinones, such as extra-mitochondrial ubiquinone (CoQ$_{10}$) or vitamin K at the expense of NAD(P)H, the NAD(P)H−CoQ$_{10}$−vitamin K−FSP1 axis averts the lipid-peroxidation chain reaction and associated ferroptosis on the level of phospholipid radicals. FSP1 is expressed in numerous cancer cell lines, and knockout of the FSP1-encoding gene, apoptosis-inducing factor mitochondria associated 2 (Aifm2), has no impact on embryo development and does not cause any overt phenotype in adult mice[10,12,13]. Therefore, FSP1 might be preferred as a target in tumor cells compared with GPX4 (refs. [14–17]),

[1]Institute of Metabolism and Cell Death, Molecular Target and Therapeutics Center, Helmholtz Munich, Neuherberg, Germany. [2]Division of Nephrology, Rheumatology and Endocrinology, Tohoku University Graduate School of Medicine, Sendai, Japan. [3]Institute of Structural Biology, Molecular Target and Therapeutics Center, Helmholtz Munich, Neuherberg, Germany. [4]transMedChem, Munich, Germany. [5]Rudolf Virchow Zentrum (RVZ), Center for Integrative and Translational Bioimaging, University of Würzburg, Würzburg, Germany. [6]Bavarian NMR Center, Department of Bioscience, School of Natural Sciences, Technical University of Munich, Garching, Germany. [7]These authors contributed equally: Bettina Proneth, Marcus Conrad. ✉e-mail: marcus.conrad@helmholtz-munich.de

which is known to be essential for early embryogenesis and tissue homeostasis in a variety of organs, such as the kidneys, liver and brain[18].

Although FSP1's central role in ferroptosis prevention has been established, detailed molecular and structural mechanisms underlying the process have remained elusive, and studies investigating mutations of FSP1 occurring in certain cancer types are still in their infancy. The first reported inhibitor for FSP1 (that is, iFSP1)[8] is specific for the human enzyme[19,20], and this specificity precludes in-depth studies on its precise mechanism of action (MoA), thus hindering the analysis of organismal differences and similarities among different FSP1 orthologs. In light of these limitations and shortcomings, we conducted comprehensive chemical and genetic screenings both to understand the molecular and structural basis of FSP1 function and FSP1's role in cellular ferroptosis vulnerability and to identify FSP1 inhibitors that can target FSP1 enzymes across different organisms.

## Results

### G244 is essential for the predicted NAD(P)H-binding site

To understand the molecular and structural basis of FSP1's functions, we surveyed consensus sequences and sequence motifs of FSP1 orthologs and nicotinamide adenine dinucleotide (NADH)–quinone reductases across various organisms. Among these, type II NADH–quinone reductases (NDH-2s) are well-characterized membrane proteins involved in the respiratory chain[21,22], and like FSP1 are flavoproteins that can efficiently reduce quinones using NADH. NDH-2s contain two conserved NADH-binding motifs (GXGXXGXE, WXXG) and one flavin adenine dinucleotide (FAD)-binding motif (GD)[23]. Interestingly, these consensus amino acid sequences in FSP1 are well-conserved among species (Fig. 1a,b and Extended Data Fig. 1a,b). Previously, the E156A and D285N substitutions, located within the GXGXXGXE and GD binding sites of human FSP1, respectively, have been reported to affect the enzymatic activity of FSP1 (refs. 9,24), although the functional relevance of G244 (WXXG motif) in FSP1 has remained unclear. In addition, a substitution (that is, p.G244D) within the NAD-binding motif is registered in the database of somatic mutations in human cancer (https://cancer.sanger.ac.uk/cosmic)[25] (Extended Data Fig. 2a,b). Thus, we first analyzed whether substituting G244 with aspartic acid affects the enzymatic activity of FSP1 in vitro[8,10]. The G244D substitution in fact abrogates its oxidoreductase activity to a much greater extent than does the E156A substitution (Fig. 1c,d). This loss in enzyme activity is very similar to that of the inactive D285N mutant, which lacks the prosthetic group, flavin adenine dinucleotide (FAD) (Fig. 1c,d). We then investigated the cellular relevance of these substitutions in ferroptosis suppression using mouse embryonic fibroblast cells with 4-hydroxytamoxifen (Tam)-inducible deletion of *Gpx4* (that is, Pfa1 cells)[26], and in cells treated with the GPX4 inhibitor (1*S*,3*R*)-RSL3 (RSL3)[2]. We included cells with the somatic substitutions V148L and S187F, which are also reported in the database, as a direct comparison, as well as other well-studied alterations such as G2A and Lyn11 membrane-anchoring sequence (Lyn11)-tagged G2A (Extended Data Fig. 2c–f). The G2A variant is known to be the myristoylation-defective mutant and strongly affects FSP1 localization, thereby abrogating its ferroptosis-suppressive function[8]. By contrast, the Lyn11-G2A mutant has a membrane-targeting sequence attached to the N-terminus of FSP1-G2A, and therefore can suppress ferroptosis[9]. Although the other reported somatic substitutions (V148L and S187F) and the Lyn11-G2A mutant enable FSP1 to protect against ferroptosis to some extent, the G244D mutant showed complete loss of its ferroptosis-suppressive function in the same way as the D285N and G2A variants (Fig.1e,f and Extended Data Fig. 2d,e). To further validate these consensus motifs, we took advantage of the protein structure of FSP1 predicted by AlphaFold2 (refs. 27,28), which perfectly aligns with the crystal structure of the yeast NDH-2 enzyme known as Ndi1 (ref. 29) (Extended Data Fig. 1b), resulting in the superimposed, modeled structure of FSP1 with putative binding sites for FAD, NADH and CoQ$_5$ (Extended Data Fig. 2g)[19]. According to this modeled FSP1

structure, G244 is an essential constituent of the NADH-binding domain (Fig. 1g), similar to other NDH-2s (Extended Data Fig. 1b), further supporting the findings that the somatic substitution G244D abrogates the ferroptosis-suppressive role of FSP1.

### Genetic screens unravel the proton-transfer function of FSP1

To gain further insights into the molecular mechanisms underlying how FSP1 reduces quinones at the expense of NAD(P)H, we established a random mutagenesis screen as an unbiased approach (Fig. 2a). Using error-prone PCR, up to three DNA mutations, on average, were introduced into the FSP1 gene *AIFM2,* resulting in one to three amino acid substitutions per FSP1 molecule. The resulting pool of genetic material was subsequently cloned into a lentivirus-based expression vector using seamless cloning. This plasmid pool was then transduced into Pfa1 cells with a very low infection ratio (the multiplicity of infection was approximately 0.1), according to genome-wide CRISPR screening approaches performed by our group and others[30–32], allowing each cell to express one FSP1 mutant. Transduced cells were divided into two groups: one was collected immediately after blasticidin selection, and the other was treated with 500 nM RSL3 for 2 d before collection. After cells were collected, genomic DNA was extracted and *AIFM2* was amplified by high-fidelity PCR and subjected to next-generation sequencing (NGS), with the goal of identifying possible mutations. Then, the complete sequence of *AIFM2* was analyzed, and the RSL3-treated group was compared with the control group. By calculating the *Z* score of each DNA sequence and annotating amino acid alterations in FSP1, we retrieved several expected loss-of function mutations that corresponded to residues in the myristoylation motif (MGXXXS)[33], the NADH-binding motif (GXGXXGXE, WXXG) and the FAD-binding motif (GD) (Fig. 2b), substantiating the validity of our approach. To further validate the screening results, we cloned the majority of detected mutations and stably expressed them in Pfa1 cells. Indeed, several mutants are sensitive to RSL3- and tamoxifen-induced ferroptosis (Fig. 2c–e and Extended Data Fig. 3a), indicating that they fail to confer the ferroptosis-suppressive function of FSP1. However, without a three-dimensional (3D) structure of FSP1, we cannot formally exclude that a fraction of these mutations not only directly impact the enzymatic activity, but also affect the binding affinity of substrates, expression levels in cells and/or proper protein folding and localization. Nonetheless, to further investigate the role of the aforementioned amino acid residues, we again took advantage of the modeled structure of FSP1 (Extended Data Fig. 2g). According to this model, D41 and C286, located in close proximity to FAD or the predicted GD motif, respectively (Figs. 1b and 2b), are suggested to be important for FAD binding (Fig. 2f). Additionally, our mutagenesis screen identified E160 and K355 as critical amino acid residues for enzyme activity of ferroptosis suppression (Fig. 2b–e). Mechanistically, NDH-2s and other members of the two-dinucleotide binding domains flavoprotein (tDBDF) superfamily, such as dihydrolipoamide dehydrogenases (DLDs), coenzyme A–glutathione reductases (GRs), apoptosis-inducing factor mitochondria associated 1 (AIFM1), ferredoxin reductases (FNRs), nitrite reductases (NIRs), NADH peroxidases and sulfide-quinone oxidoreductases (SQRs), have the glutamic acid residue (Glu) as a catalytic site, like E172 in *Staphylococcus aureus* (Fig. 2g), as well as a nearby glutamic acid and/or lysine, like K379 in *S. aureus*[34]. In line with this mechanism, E156 and K355 of FSP1 are structurally well conserved. Additionally, sequential acidic amino acid residues in close proximity to the glutamic acid have been proposed to be part of a conserved proton-transfer function during quinone reduction[23]. This proton-transfer function presumably consists of a proton transfer through sequential, conserved carboxylic residues in the α-helix (that is E172, E176, D179 and E183 for *S. aureus* (Fig. 2g), E169, E173, D176 and E180 for *C. thermarum*, and E242, E246, D249 and D254 for *S. cerevisiae*, respectively), and a residue for hydrogen bond formation and subsequent protonation of the quinone upon reduction in the active site (that is, K379 for *S. aureus*, K376 for *C. thermarum*,

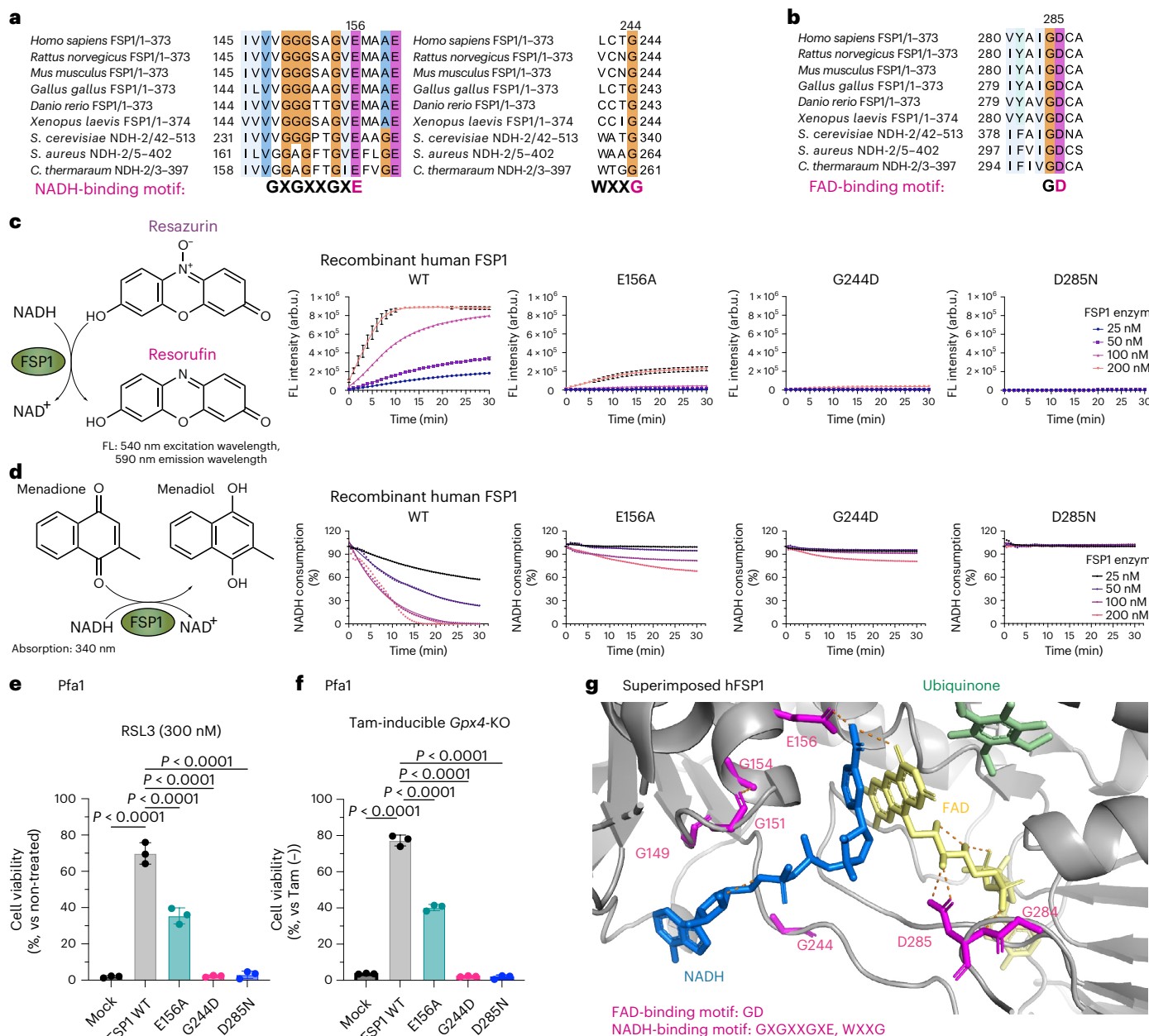

**Fig. 1 | G244 is essential for the predicted NAD(P)H-binding site of FSP1.**
**a**, Conserved NADH-binding site (GXGXXGXE and WXXG). Protein alignment of FSP1 from different species: *Homo sapiens* (human), *Mus musculus* (mouse), *Rattus norvegicus* (rat), *Gallus gallus* (chicken), *Xenopus laevis* (frog), *Danio rerio* (zebrafish), *Saccharomyces cerevisiae* Ndi1 (PDB: 4G73), *Caldalkalibacillus thermarum* NDH-2 (PDB: 5NA1), and *Staphylococcus aureus* NDH-2 (PDB: 4NWZ). The numbers next to the organisms indicate the range of the amino acids of each gene used for generating the protein alignment. **b**, Conserved FAD-binding site (GD). Protein alignment of FSP1 from different species. **c**, Schematic representation of the FSP1 enzyme activity assay with resazurin used as the substrate (left). Reduction of resazurin in the presence of wild-type (WT) FSP1, FSP1-E156A, FSP1-G244D, or FSP1-D285N at the indicated concentrations. Data are shown as the mean ± s.d. of 3 wells of a 96-well plate, from 1 of 3 independent experiments. FL, fluorescence. **d**, Schematic representation of FSP1 enzyme activity assay, with menadione used as the substrate (left). Oxidation of NADH in the presence of WT FSP1, FSP1-E156A, FSP1-G244D, or FSP1-D285N at the

indicated concentrations. Data are from a single well of a 96-well plate from 1 of 3 independent experiments. **e**, Viability of Pfa1 cells stably overexpressing WT HA-tagged human FSP1 (hFSP1-HA) or FSP1 mutants, treated with 300 nM RSL3 for 24 h. Data are shown as the mean ± s.d. of 3 wells of a 96-well plate from 1 of 3 independent experiments. **f**, Viability was measured in Pfa1 cells stably overexpressing WT hFSP1 or one of the mutants with or without treatment with Tam (1 μM) for 72 h. Data were normalized to each group that was not treated with Tam. Data are shown as the mean ± s.d. of 3 wells of a 96-well plate from 1 of 3 independent experiments. *P* values were calculated using a one-way analysis of variance (ANOVA) followed by Dunnett's multiple-comparison test (**e**,**f**). **g**, Predicted FSP1 structure, with consensus NADH- and FAD-binding motifs, from the AlphaFold2 database (https://alphafold.ebi.ac.uk). The co-factors, FAD (yellow), NADH (blue) and CoQ₅ (green), were embedded from the structure of the yeast ortholog, NDH-2 (Ndi1) (PDB: 4G73). The expected hydrogen bond was generated by Pymol.

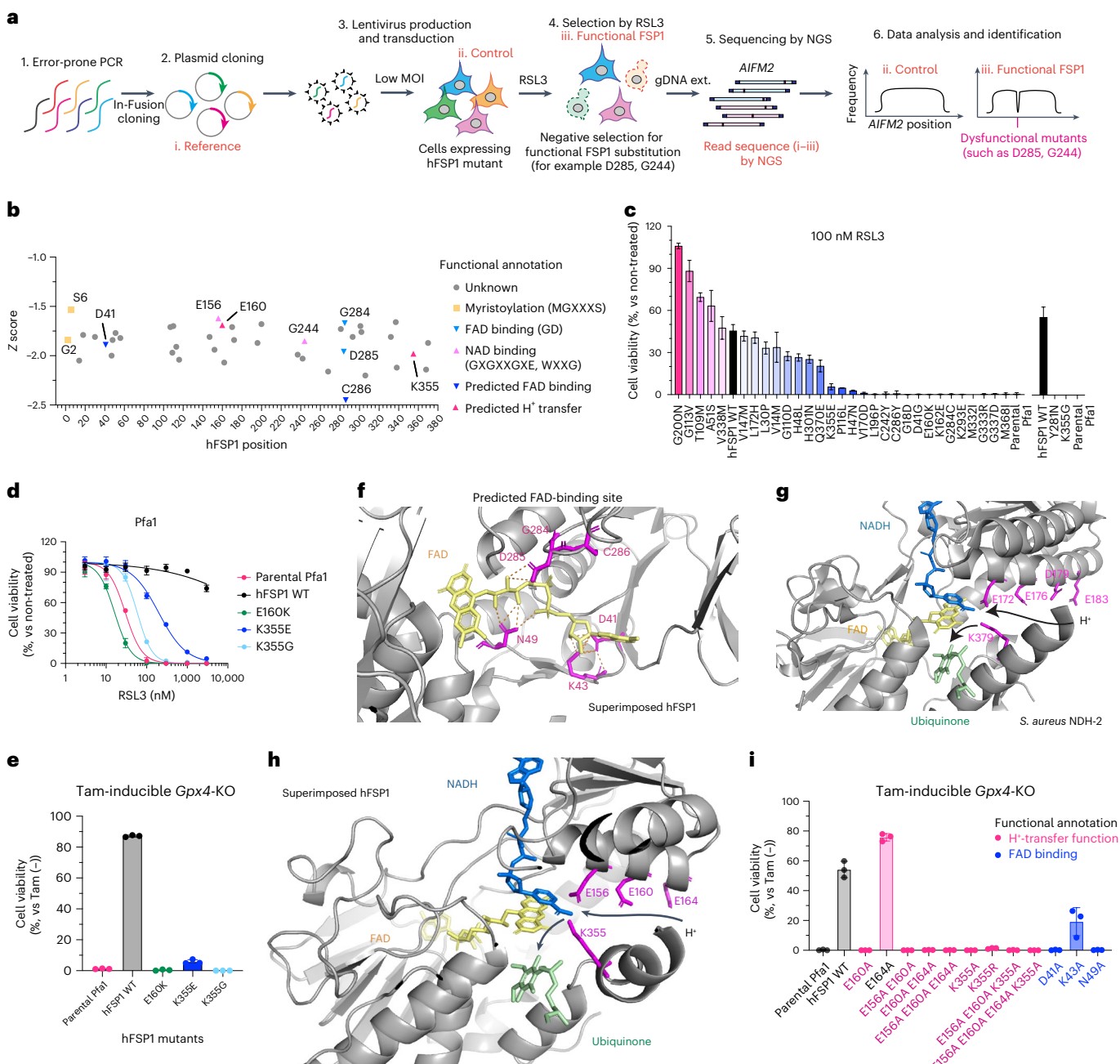

**Fig. 2 | An unbiased genetic screen uncovers the proton-transfer function of FSP1. a**, Schematic of the mutational screen. *AIFM2* was mutated using error-prone PCR (Step 1). PCR fragments were then cloned into plasmids (i. Reference) using the seamless cloning enzyme "in-Fusion" (Step 2). Lentivirus pools using randomly mutated plasmids were transduced into Pfa1 cells (ii. Control) with low multiplicity of infection (MOI) (Step 3). Cells overexpressing mutated FSP1 were selected by RSL3 (iii. Functional FSP1) (Step 4). Surviving clones were harvested and genomic DNA was extracted and followed by NGS sequencing (Step 5). Sequencing results were analyzed and mutations were identified (Step 6). gDNA ext., genomic DNA extraction. **b**, Representative summary of dysfunctional FSP1 substitutions after *Z*-score calculation. Mutated residues in the myristoylation motif (MGXXXS, yellow), NADH-binding motif (GXGXXGXE, WXXG, light pink) or FAD-binding motif (GD, light blue), and those predicted to affect proton-transfer function (E160 and K355, pink) or FAD binding (D41 and C286, blue), are shown. **c**, Viability of Pfa1 cells stably overexpressing hFSP1-HA or mutant FSP1 that were treated with 100 nM RSL3 for 24 h or that were not treated. Data are shown as the mean ± s.d. of 3 wells of a 384-well plate (left) or a 96-well plate (right) from one experiment. **d**, Viability of Pfa1 cells stably overexpressing WT hFSP1-HA or mutant FSP1 that were treated with RSL3 for 24 h or that were not treated.

**e**, Viability of Pfa1 cells stably overexpressing WT hFSP1 or a mutant that were treated with 1 µM Tam for 72 h or that were not treated. Data were normalized to each group that was not treated with Tam (Tam (−)). Data are shown as the mean ± s.d. of 3 wells of a 96-well plate from 1 of 3 independent experiments (**d,e**). **f**, Superimposed hFSP1 structure with FAD (yellow) and FAD-binding residues. FAD was embedded from the structure of *S. aureus* NDH-2 (PDB: 5NA1), and expected hydrogen bonds were generated using Pymol. **g**, Superimposed *S. aureus* NDH-2 structure (PDB: 5NA1) with the co-factors FAD (yellow), NADH (blue) and CoQ₅ (green). The proton transfer through sequential carboxylic residues in the α-helix (that is, E172, E176, D179 and E183) and subsequent final protonation from K379 to ubiquinone are indicated with black arrows. **h**, Proposed FSP1 proton transfer through sequential carboxylic residues in the α-helix (that is, E156, E160 and E164) and subsequent final protonation from K355 to ubiquinone are indicated with black arrows. **i**. Viability was measured in Pfa1 cells stably overexpressing WT hFSP1 or mutants after treatment with 1 µM Tam for 72 h or without treatment. Data were normalized to each group that was not treated with Tam. Data are shown as the mean ± s.d. of 3 wells of a 96-well plate from 1 of 2 independent experiments.

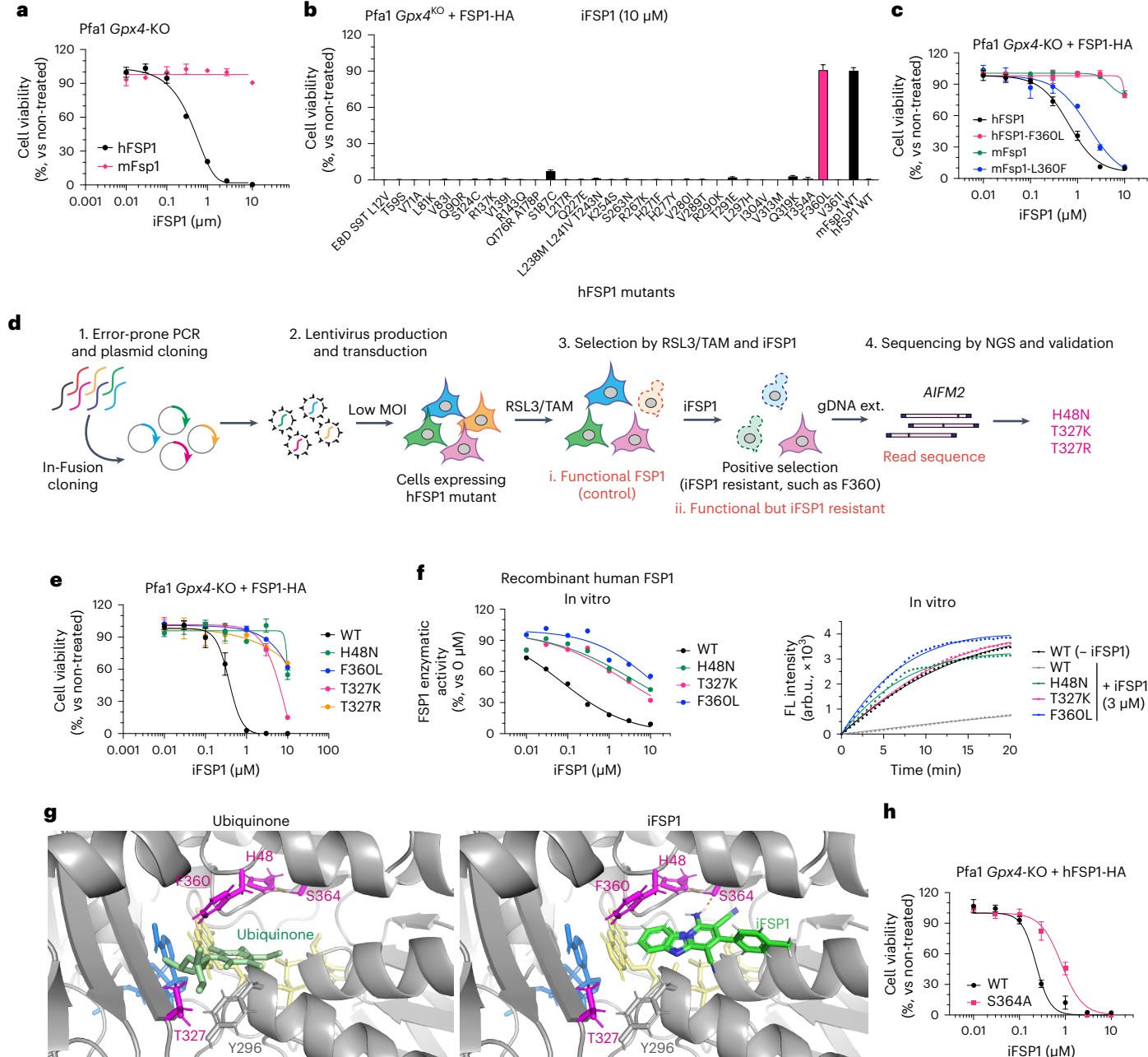

**Fig. 3 | iFSP1 targets the quinone-binding pocket. a**, Viability of Pfa1 *Gpx4*-KO cells stably overexpressing hFSP1-HA or mouse (mFsp1)-HA treated with iFSP1 for 24 h. **b**, Viability of Pfa1 *Gpx4*-KO cells stably overexpressing WT FSP1-HA or an FSP1 mutant, treated with 10 μM iFSP1 for 24 h. **c**, Viability of Pfa1 *Gpx4*-KO cells stably expressing hFSP1, hFSP1-F360L, mFsp1, or mFsp1-L360F, treated with iFSP1 for 24 h. **d**, Schematic representation of the mutational screen. **e**, Viability of Pfa1 *Gpx4*-KO cells stably expressing WT FSP1-HA or an FSP1 mutant, treated with iFSP1 for 24 h. **f**, Representative dose–response curves for the effect of iFSP1 on the activity of FSP1 mutants, using recombinant purified hFSP1 protein (left). Representative reaction curve for the effect of 3 μM iFSP1 or 0 μM iFSP1 on the activity of WT FSP1 and the mutant variants (right). Data are from a single well of a 96-well plate from 1 of 3 independent experiments. arb.u., arbitrary units. **g**, Comparison between iFSP1 and quinone in the binding pocket. H48, T327, S364 and F360 are highlighted in magenta. In silico simulation identified the interaction between S364 and iFSP1 using a hydrogen bond, and H48 is expected to stabilize its orientation. Each interaction is depicted by the dashed line in orange. **h**, Cell viability of Pfa1 *Gpx4*-KO cells stably overexpressing WT hFSP1 or hFSP1-S364A, treated with iFSP1 for 24 h. Data are shown as the mean ± s.d. of 3 wells of a 96-well plate from 1 of 3 independent experiments (**a–c**,**e**,**h**).

and Y482 for *S. cerevisiae*, respectively) (Fig. 2g and Extended Data Fig. 3b,c). In accordance with this protonation pathway, E156, E160 and E164 are present in the α-helix, and K355 is in close proximity to the catalytic site in FSP1 (Fig. 2h). Thus, this highly conserved proton-transfer mechanism involving sequential carboxylic acid and lysine residues is most likely crucial for the quinone protonation function of FSP1. Therefore, amino acid mutations identified by random

mutagenesis, namely E160K, K355E and K355G, can be considered detrimental because they potentially interrupt the proton-transfer chain. To validate these FAD and proton-transfer functions, we generated corresponding alanine mutants and overexpressed them in Pfa1 cells. Except for the E164A and K43A mutants, all remaining Ala substitutions failed to rescue from ferroptosis induced by either knockout (KO) of *Gpx4* or pharmacological inhibition of GPX4,

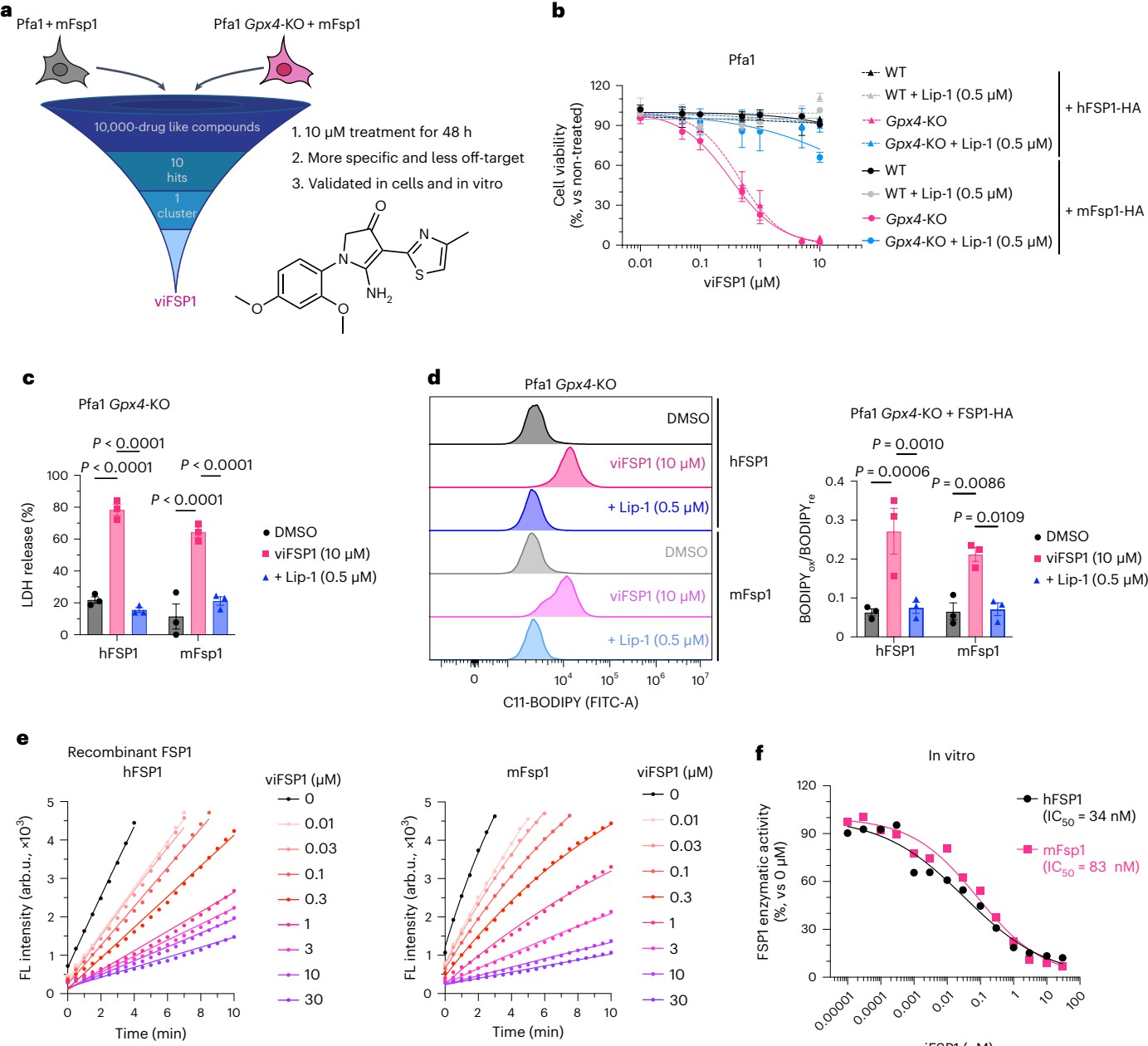

**Fig. 4 | Identification of viFSP1 as a species-independent FSP1 inhibitor.**
**a**, Schematic of the compound screen to identify new mFsp1 inhibitors, and the chemical structure of viFSP1. Survival of Pfa1 *Gpx4*-KO + mFsp1 cells relies on mFsp1 function. **b**, Viability of Pfa1 *Gpx4*-WT and *Gpx4*-KO cells stably overexpressing human or mouse FSP1-HA, treated with viFSP1 for 24 h. Lip-1 was used to inhibit ferroptosis. **c**, Lactate dehydrogenase (LDH) release was determined after treating Pfa1 *Gpx4*-KO cells overexpressing hFSP1-HA or mFsp1-HA with DMSO, 10 µM viFSP1 or 0.5 µM Lip-1 for 24 h. **d**, Lipid peroxidation was evaluated by C11-BODIPY 581/591 staining after treating *Gpx4*-KO cells stably overexpressing hFSP1-HA or mFsp1-HA with DMSO, 10 µM viFSP1 or 0.5 µM Lip-1

for 3 h. The plot is representative of three independent experiments (left), and quantified median values of three independent experiments (right) are shown. BODIPY$_{ox}$, fluorescence of oxidized BODIPY. BODIPY$_{re}$, fluorescence of reduced BODIPY. Data are shown as the mean ± s.e.m. of three independent experiments (**b**–**d**). *P* values were calculated using a one-way ANOVA followed by Tukey's multiple-comparison test (**c**,**d**). **e**, Representative assays of the inhibitory effect of different concentrations of viFSP1 on hFSP1 (left) or mFsp1 (right), using recombinant purified proteins. **f**, Representative dose–response curves for the effect of viFSP1 on hFSP1 and mFsp1 activity. Data are from a single well of a 96-well plate from 1 of 2 independent experiments (**e**,**f**).

suggesting that these positions are essential for FSP1 function (Fig. 2i and Extended Data Fig. 3d,e).

## iFSP1 targets the quinone-binding pocket

We previously discovered iFSP1, the first FSP1 inhibitor[8], although the mode of inhibition and the binding site remained unknown. First, we focused on the inherent differences in half-maximal inhibitory concentration (IC$_{50}$) values between human and mouse FSP1 isoforms, because

iFSP1 is specific for the human protein[19,20]. Indeed, Pfa1 *Gpx4*-KO cells stably overexpressing human FSP1 are sensitive to iFSP1 treatment, whereas mouse-Fsp1-expressing cells are resistant to it (Fig. 3a). To identify the possible amino acids residue or residues that is or are targeted by iFSP1, we first site-directly altered all human FSP1 residues to their mouse Fsp1 counterparts in a step-by-step fashion and confirmed that their expression levels were sufficient to confer potential resistance to ferroptosis by Tam-inducible KO of *Gpx4* in Pfa1 cells (Extended Data

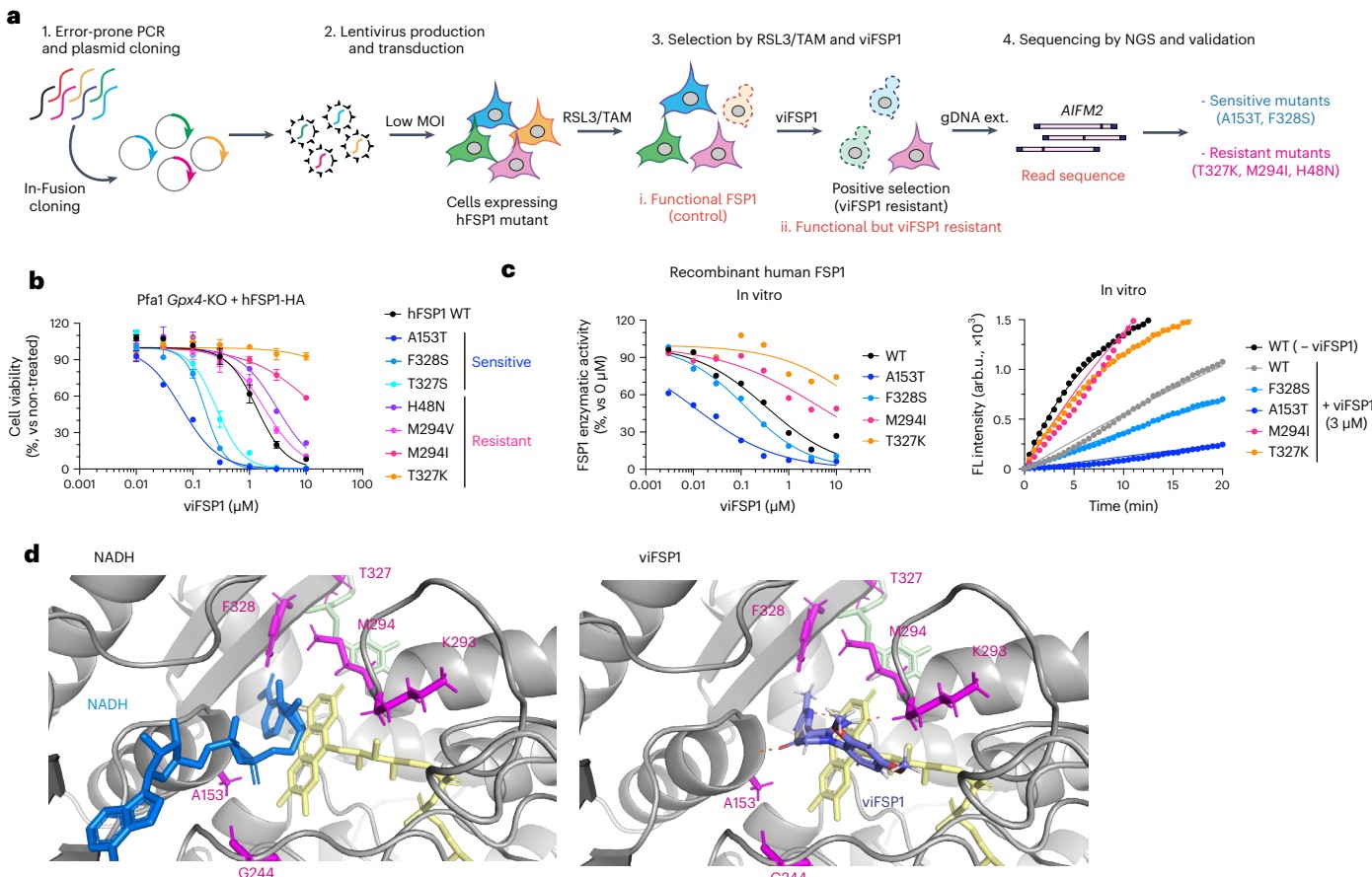

**Fig. 5 | viFSP1 targets the NAD(P)H-binding pocket of FSP1. a**, Schematic of the mutational screen to identify the responsible sites affecting viFSP1 activity in hFSP1. **b**, Viability in Pfa1 *Gpx4*-KO cells stably overexpressing mutant hFSP1-HA, treated with viFSP1 for 24 h. Data are shown as the mean ± s.d. of 3 wells of a 96-well plate from 1 of 3 independent experiments (See also Supplementary Videos 1–4). **c**, Representative dose–response curves for the effect of viFSP1 on the activity of WT FSP1 or its mutants, using recombinant purified hFSP1 protein (left). Representative in vitro assays of inhibition of WT FSP1 and the mutant variants by 3 μM viFSP1 or 0 μM viFSP1 (right). Data are from a single well of a 96-well plate from 1 of 3 independent experiments. **d**, Comparison between viFSP1 and NADH in the respective binding pockets. A153, G244, M294, T327 and F328 are highlighted in magenta. In silico simulation identified the interaction between the main chains of A153, K293 and viFSP1 using a hydrogen bond. Each interaction is depicted by the dashed line in orange.

Fig. 4a,b). Cells stably expressing the mutated human FSP1 enzyme were then treated with iFSP1 for 24 h. Surprisingly, the F360L mutant emerged to be as resistant to iFSP1 treatment as was murine FSP1, and vice versa in murine Fsp1 (Fig. 3b,c)[20]. Thus, we hypothesized that F360 and its surrounding residues are part of the putative binding pocket of iFSP1. To further pinpoint the actual binding site, we performed a random PCR mutagenesis screen to identify iFSP1-resistant mutants (Fig. 3d). Similar to the RSL3 screening method described above (Fig. 2a), the randomly mutated FSP1 pool was overexpressed in Pfa1 cells and selected with RSL3 for 2 d. Afterwards, cells were treated with Tam to remove dysfunctional FSP1 mutants. Then, TAM/RSL3-selected cells were treated for 7 d with iFSP1 to establish iFSP1-resistant clones, followed by NGS analysis. After data analysis and validation, we identified H48N, T327K and T327R as iFSP1-resistant mutants. For validation, the viability of Pfa1 *Gpx4*-KO cells expressing F360L, H48N, T327K or T327R was determined 24 h after iFSP1 treatment, which showed that they indeed are all resistant to iFSP1 (Fig. 3e and Extended Data Fig. 4c).

To confirm that iFSP1 directly inhibits human FSP1 and does not act up- or downstream in the NAD(P)H–CoQ$_{10}$–vitamin K–FSP1 axis, we heterologously expressed FSP1 harboring either the H48N, T327K or F360L substitution in *Escherichia coli* and purified recombinant FSP1 variants. We found that, when tested in the classical FSP1 enzyme assay, F360L, H48N and T327K mutants are resistant to iFSP1, corroborating that iFSP1 is a direct inhibitor of human FSP1 (Fig. 3f and Extended Data

Fig. 4d). Interestingly, human, *Gallus gallus* (Chicken), and *Xenopus laevis* (frog) all contain a phenylalanine at position 360 (human FSP1 numbering), whereas mouse, rat (*Rattus norvegicus*) and *Danio rerio* (zebrafish) have a leucine at the corresponding position (Extended Data Fig. 4e). Moreover, human FSP1-T327S, which reflects the serine residue found in *X. laevis*, is also sensitive to iFSP1 (Extended Data Fig. 4f). Thus, we selected and examined F360 (human and chicken) and L360 (mouse and rat), and, as expected, Pfa1 *Gpx4*-KO cells overexpressing human or chicken FSP1 were sensitive to iFSP1, whereas those expressing rat or mouse FSP1 were resistant to iFSP1 (Extended Data Fig. 4g,h). In light of the fact that these mutants face the expected membrane-attaching surface and quinone-binding pocket (Extended Data Fig. 2g), it can be assumed that iFSP1 targets the quinone-binding site (Fig. 3g)[19,20]. According to this model, iFSP1 would bind to FSP1 via the hydrogen bond between S364 and iFSP1 and the π–π interaction between F360, Y296 and iFSP1 (Fig. 3g,h).

## viFSP1 is a species-independent FSP1 inhibitor

iFSP1 is not applicable for FSP1-inhibition studies in experimental rodent models (Extended Data Fig. 4g). Thus, to better understand the role of FSP1 in a broader context, including species-specific differences, inhibitors targeting FSP1 in species besides humans are highly warranted. To this end, we screened a library of 10,000 drug-like small molecules in murine-FSP1-expressing Pfa1 cells (Fig. 4a) and identified a

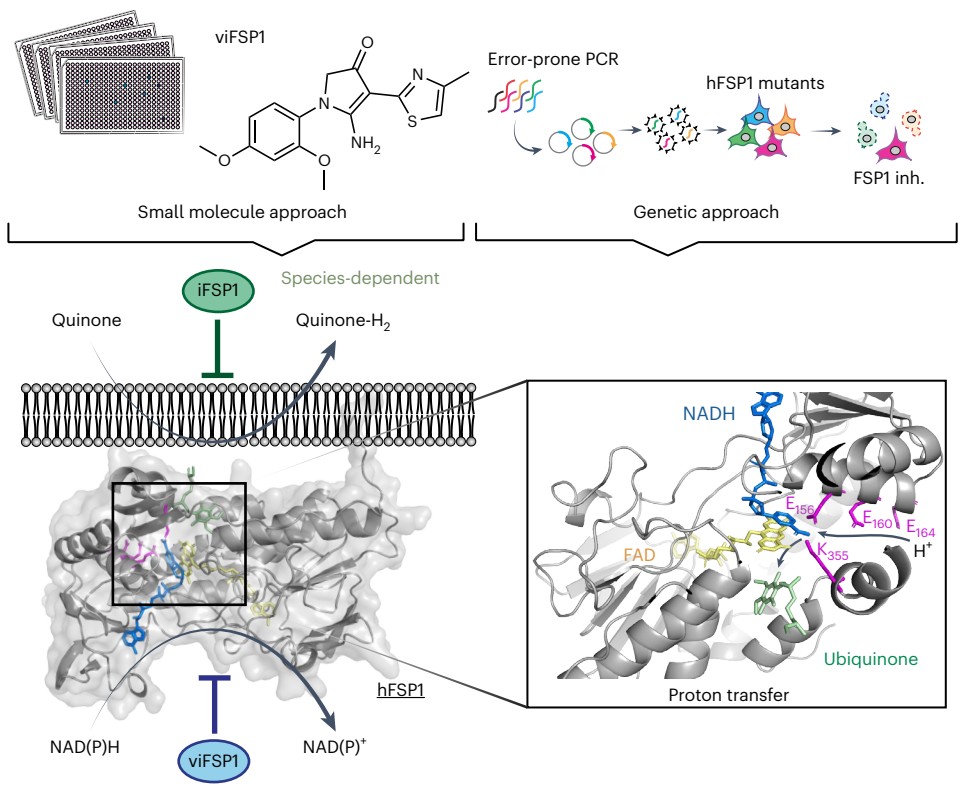

**Fig. 6 | Mechanistic insights into FSP1 and its different target sites using FSP1 inhibitors.** The chemical approach using the compound library and the genetic approach using targeted and untargeted mutagenesis uncover the mechanism of action of two representative FSP1 inhibitors. Human-specific iFSP1 targets the quinone-binding pocket, whereas species-independent viFSP1 targets the NAD(P)H-binding pocket. Besides, a well-conserved proton transfer function is critical for the anti-ferroptotic activity of FSP1.

compound, referred to hereafter as versatile inhibitor of FSP1 (viFSP1), that is a new inhibitor of human and murine FSP1. Treatment with viFSP1 causes marked lipid peroxidation and associated ferroptotic cell death in Pfa1 cells stably expressing either murine Fsp1 or human FSP1 (Fig. 4b–d). viFSP1-induced cell death was rescued by the ferroptosis inhibitor liproxstatin-1 (Lip-1)[35], confirming that viFSP1 indeed causes ferroptotic cell death in cells whose survival depends on FSP1 (Extended Data Fig. 5a). To address whether viFSP1 may also kill (cancer) cells by targeting FSP1, co-treatment of viFSP1 with pharmacological inhibitors of GPX4 (RSL3 (ref. 36) and JKE-1674 (ref. 37)) or genetic deletion of GPX4 synergistically[38,39] induced ferroptosis in a number of human and murine cancer cell lines as well as rat fibroblast (Rat1) and Pfa1 cells overexpressing FSP1 from different species (Extended Data Fig. 5b–h and Extended Data Fig. 6a–j). To perform an initial structure–activity relationship study (SAR), we obtained a handful of commercially available analogs of viFSP1 in addition to custom-made ones, indicating that the substitution pattern of methoxy-groups around the phenyl ring influences human/mouse selectivity. The compound structures and corresponding half-maximal effective concentration ($EC_{50}$) values are depicted in Extended Data Figure 5c. We then studied whether this next-generation FSP1 inhibitor acts as a direct inhibitor of FSP1 using recombinant hFSP1 and mFsp1 enzymes in the resazurin reduction assay[8,10] (Fig. 4e,f). Indeed, both hFSP1 and mFsp1 enzyme activity are equally sensitive to viFSP1 in a cell-free target-based system, and calculated $IC_{50}$ values do not show a major difference among human and murine enzymes (Fig. 4f). Thus, this set of data strongly indicates that viFSP1 is a direct species-independent FSP1 inhibitor.

To further investigate the competitive mechanism of action of viFSP1, the relationship of FSP1 enzymatic activity and its substrates was analyzed using an in vitro assay. Taking advantage of the fact that

FSP1 can also bind and reduce resazurin at the expense of NADH[8], we calculated the velocity of the enzyme-catalyzed reaction at infinite concentration of substrate ($V_{max}$) and the Michaelis constant ($K_m$) in the presence of FSP1 inhibitors using increasing amounts of either NADH or resazurin (Extended Data Figs. 4i and 7a). From the Lineweaver–Burk and Dixon plots[40], it can be speculated that iFSP1 and viFSP1 are both non-competitive inhibitors, which means that these FSP1 inhibitors can bind the enzyme in the presence or absence of a substrate; however, this should be investigated experimentally when the 3D structure of FSP1 becomes available.

**viFSP1 targets the NAD(P)H-binding pocket of FSP1**

To understand the inhibitory mechanism of viFSP1 in more detail, we performed a random mutagenesis screen to identify a viFSP1-resistant mutant version of FSP1 (Fig. 5a). Akin to the iFSP1 screen described above (Fig. 3), Pfa1 *Gpx4*-KO cells stably overexpressing randomly mutated yet functional FSP1 enzymes were selected by viFSP1 for 7 d, followed by NGS analysis. After data analysis and validation, we identified A153T and F328S as viFSP1-sensitive mutants, and H48N, M294I, M294V, T327K and T327R as viFSP1-resistant mutants (Fig. 5b and Extended Data Figs. 4c and 7b). Pfa1 *Gpx4*-KO cells overexpressing A153T or F328S were at least ten times more sensitive to viFSP1 treatment than were cells overexpressing wild-type FSP1; cells overexpressing the H48N, M294I, M294V, T327K or T327R variants were approximately five to ten times more resistant to viFSP1 treatment (Fig. 5b). In addition to the T327K variant of FSP1, the A153T, F328S and M294I variants were examined in the in vitro FSP1 enzyme assay using purified recombinant human FSP1. Thereby, we could observe the same tendency for FSP1 inhibition in the cell-free system using viFSP1 as in cell culture, showing that viFSP1 is a direct inhibitor of FSP1

(Fig. 5c and Extended Data Fig. 7b). Because these amino acids are located in the NAD(P)H-binding site and are highly conserved among species (Extended Data Fig. 7c–f), it can be assumed that viFSP1 targets the NAD(P)H-binding pocket (Fig. 5d).

## Discussion

Using a series of targeted and untargeted chemical–genetic screens, we report here unprecedented insights into contributions of the NAD(P)H-binding motif and the proton-transfer function to FSP1 activity (Fig. 6), in addition to the identification of a species-independent FSP1 inhibitor, viFSP1. A number of point mutations in the *AIFM2* gene are present as somatic mutations in people with cancer, including those encoding p.G244D, p.E160D, pE160 stop, p.K355R and p.D285N (Extended Data Fig. 8a). In light of the loss of function of these mutants, cancer cells harboring these somatic mutations (heterozygous mutations) might be more vulnerable to ferroptosis. Given that some cancer cells are resistant to GPX4-inhibition-induced ferroptosis, and that FSP1 inhibitors sensitize a number of cancer cells to ferroptosis induced by sublethal GPX4 inhibition, the combination therapy of FSP1 inhibitors with canonical ferroptosis inducers, such as GPX4 and system $x_c^-$ inhibitors, ideally in a tumor-specific manner, could potentially be a new anticancer therapy. Depmap analysis (https://depmap.org/portal/, v23Q2) revealed that ovarian cancer cells (SKOV3 cells) and endometrial/uterine cancer cells (RL952 cells) harbor G337D and S6L substitutions, respectively, in FSP1 Extended Data Fig. 8b,c). Considering that S6 is part of the myristoylation consensus motif (Fig. 2b)[33], and that G337D fails to protect against ferroptosis (Fig. 2c), these cancer cell lines might be more vulnerable to ferroptosis induction.

Although iFSP1 was reported as the first FSP1 inhibitor[8] and it was predicted to target the quinone-binding site in an in silico simulation[19], its MoA, including a potential binding pocket, has remained obscure. The combination of hypothesis-driven and unbiased approaches using site-directed and error-prone PCR mutagenesis, respectively, has now enabled us to pinpoint H48, F360 and T327 as being necessary for iFSP to bind and inhibit human FSP1, which was also reported in a recent independent study[20]. Considering that some of these amino acid residues can differ substantially among species, this knowledge could be used to develop FSP1 inhibitors to achieve species-specific selectivity. Moreover, bioinformatic analyses of people with cancer harboring T327M mutations in FSP1 (Extended Data Fig. 6b)[25] suggest that certain individuals might be resistant to treatment with FSP1 inhibitors, thus likely necessitating personalized treatment approaches.

Because iFSP1 is specific for human FSP1, we introduce here a species-independent FSP1 inhibitor, viFSP1, that will prove useful when studying the functions of FSP1 orthologs. Notably, FSEN15 (ref. 41), a compound based on a similar scaffold, was recently reported by another group, although neither a detailed SAR analysis nor a comprehensive characterization of the interaction site have been presented. In any case, we now characterize viFSP1, which is by far more potent in cultured cells and in a cell-free system, as a direct inhibitor targeting the NADH-binding pocket surrounding residues A153, F328, M294 and T327 of FSP1.

By using random mutagenesis screens with error-prone PCR to identify the binding pocket of FSP1 inhibitors, we infer that the screening strategy described herein can be applied to other compounds when searching for the binding pocket of related target proteins[42]. Thanks to recent advances in artificial-intelligence technologies, and when combined with the predicted structures from AlphaFold2 and other docking simulations, the identification of pharmacological inhibitors will become more accurate, straightforward and reliable. We confirmed that all mutants that are resistant or sensitive to FSP1 inhibitors indeed afford survival of cells after genetic deletion of *Gpx4*, suggesting that these mutants must be functional. However, as experimental data on the 3D structure of FSP1 remain elusive at this stage, we cannot formally exclude the possibility that amino acid alterations can affect folding and/or other unrecognized post-translational modifications of FSP1. As shown above, this kind of information can be very useful; however, caution must be taken when interpreting data on the basis of modeling approaches. An experimentally validated 3D structure of FSP1 is needed in the near future. As a showcase, we applied these genetics and bioinformatics approaches to FSP1 and identified a series of mutations, which ultimately dictate the sensitivity of cells to ferroptosis. The information collected from the mutational analysis might prove highly beneficial in pharmacogenetic studies to predict the efficacy of ferroptosis therapy for each individual patient in clinical settings in the future.

## Online content

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

## Methods

### Chemicals

Liproxstatin-1 (Lip-1: Selleckchem, cat. no. S7699), (1S,3R)-RSL3 (RSL3: Cayman, cat. no.19288), iFSP1 (ChemDiv, cat. no. 8009-2626 or Cayman cat. no. Cay29483), viFSP1 (ChemDiv, cat. no. D715-1847 or Vitas M Laboratory, cat. no. STK626779), deferoxamine mesylate salt (DFO: Sigma, cat. no. 138-14-7), ferrostatin-1 (Fer-1: Sigma, cat. no. SML0583), zVAD-FMK (zVAD: Enzo Life Sciences, cat. no. ALX-260-02), necrostatin-1s (Nec-1s: Enzo Life Sciences, cat. no. BV-2263-5), MCC950 (Sigma, cat. no. 5381200001), olaparib (Selleckchem, cat. no. S1060) and JKE-1674 (Cayman, cat. no. Cay30784-1) were used in this study. Custom-made compounds were obtained from Intonation Research Laboratories.

### Cell lines

4-hydroxytamoxifen (Tam)-inducible $Gpx4^{-/-}$ murine immortalized fibroblasts (referred to as Pfa1 cells) have been reported previously[26]. Genomic $Gpx4$ deletion can be achieved using Tam-inducible Cre recombinase and the CreER$^{T2}$–LoxP system. HT-1080 (CCL-121), HEK293T (CRL-3216), 786-O (CRL-1932), A375 (CRL-1619), B16F10 (CRL-6475), LLC (CRL-1642), MDA-MB-436 (HTB-130), SW620 (CCL-227), NCI-H460 (HTB-177) and 4T1 (CRL-2539) cells were obtained from ATCC. SKOV3 (91091004) cells were obtained from Sigma-Aldrich. HEC151 cells (JCRB1122-A) were obtained from Tebubio. MC38 cells (available from Sigma) were a gift from P. Agostinis (KU Leuven, Belgium). Rat1 cells (available from Thermo Fisher) were a gift from Medizinische Hochschule Hannover. Huh7 cells (available from Thermo Fisher) were a gift from R. Schneider (Helmholtz Munich). Pfa1, 786-O, A375, SW620, Huh7, HT-1080, Rat1, MC38, LLC and B16F10 cells were cultured in DMEM-high glucose (4.5 g glucose L$^{-1}$) with 10% fetal bovine serum (FBS), 2 mM L-glutamine and 1% penicillin–streptomycin. MDA-MB-436, HEC151, SKOV3, H460 and 4T1 cells were cultured in RPMI GlutaMax with 10% FBS and 1% penicillin–streptomycin. To generate cell lines stably overexpressing FSP1, antibiotics (puromycin 1 µg mL$^{-1}$ and blasticidin 10 µg mL$^{-1}$) were used for selection. For culturing GPX4-deficient cells, 1 µM Lip-1 was supplemented. All cells were cultured at 37 °C with 5% $CO_2$ and verified to be negative for mycoplasma.

### Production and isolation of FSP1 enzyme

Recombinant human and mouse FSP1 proteins and FSP1 mutants were produced in BL21 $E.$ $coli$ and purified by affinity chromatography with a Ni-NTA system[8].

### FSP1 enzyme activity and inhibition assay

For the resazurin assay, enzyme reactions in TBS buffer (50 mM Tris-HCl, 150 mM NaCl) containing 15–200 nM recombinant human or mouse FSP1 and their mutants, 200 µM NADH and the inhibitors (iFSP1 and viFSP1) were prepared. After the addition of 100 µM resazurin sodium salt (Sigma, cat. no. R7017), the fluorescence intensity (FL intensity, excitation/emission wavelengths (Ex/Em) = 540 nm/590 nm) was recorded every 30 or 60 s using SpectraMax M5 or SpectraMaxiD5 microplate reader with SoftMax Pro v7 (Molecular devices) at 37 °C. Reactions without resazurin (inhibitor) were used to normalize and calculate FSP1 enzymatic activity and IC$_{50}$ values. Curve fitting and calculation of IC$_{50}$ values were performed using GraphPad Prism v9.

For the NADH-consumption assay, enzyme reactions in PBS (Gibco, cat. no.14190094) containing 25–200 nM recombinant human FSP1 (or its mutants) and 50 µM menadione (Sigma, cat. no. M5625) were prepared. After the addition of 200 µM NADH, the absorbance at 340 nm at 37 °C was measured every 30 s using SpectraMax M5 Microplate Reader (Molecular Devices). Reactions without NADH or without enzyme were used to normalize the results. Curve fitting was done using GraphPad Prism v9.

For analysis of enzyme kinetics, reactions in PBS (Gibco, cat. no. 14190094) containing 100 nM recombinant human FSP1 and 0.04–500 µM NADH with 100 µM resazurin sodium salt or 0.03–100 µM resazurin with 200 µM NADH were mixed; then, FL intensity

(Ex/Em = 540/590 nm) was recorded every 60 s using the Spectra MaxiD5 microplate reader (Molecular Devices) at 37 °C. The initial slope of FL intensity was used for following the calculations of $K_m$ and $V_{max}$ using GraphPad Prism v9.

### Cell viability assay

Cells were seeded on 96-well plates (2,000–10,000 cells per well for ferroptosis inducers and 500 cells per well for TAM) or 384-well plates (800 cells per well) and cultured overnight. The next day, the medium was changed and the following compounds were added: RSL3, iFSP1, viFSP1, Lip-1, DFO, Fer-1, zVAD, Nec-1s, MCC950 and olaparib, at the indicated concentrations in each figure. Cell viability was determined after 24–48 h (for RSL3, iFSP1, and viFSP1) or 72 h (for TAM) upon treatment, using AquaBluer (MultiTarget Pharmaceuticals, cat. no.6015) or 0.004% resazurin as an indicator of viable cells.

As readout, fluorescence was measured at Ex/Em = 540/590 nm using a SpectraMax M5 microplate reader with SoftMax Pro v7 (Molecular devices) after 4–6 h of incubation in normal cell-culture medium. Cell viability (%) was normalized and calculated using untreated conditions, which is in the absence of ferroptosis inducers or tamoxifen.

The synergistic effect was assessed using MuSyC (https://musyc.lolab.xyz)[38,39].

### LDH release assay

For the LDH release assay, 2,500 cells per well were seeded on 96-well plates and cultured overnight. On the following day, the medium was changed to the fresh DMEM containing inhibitors and incubated for another 24 h. Necrotic cell death was determined using the Cytotoxicity Detection kit (LDH) (Roche, cat. no.11644793001) following the manufacturer's protocol. In brief, cell-culture supernatant was collected as a sample of the medium, and cells were then lysed with 0.1% Triton X-100 in PBS as a lysate sample. Medium and lysate samples were individually mixed with reagents on the 96-well plate, and the reaction mixture was incubated for 15–30 min at room temperature. Then, the absorbance was measured at 492 nm using the SpectraMax M5 microplate reader (Molecular Devices). The cell death ratio was calculated by LDH release (%) as follows: (absorbance (abs) of medium sample) / ((abs of lysate) + (abs of medium samples)) × 100.

### Lipid peroxidation assay

One day before the experiments, 100,000 cells per well were seeded on a 12-well plate. On the next day, cells were treated with 10 µM viFSP1 for 3 h and were then incubated with 1.5 µM C11-BODIPY 581/591 (Invitrogen, cat. no. D3861) for 30 min in a 5% $CO_2$ atmosphere at 37 °C. Subsequently, cells were washed with PBS once and trypsinized at 37 °C. Then, cells were resuspended in 500 µL PBS and passed through a 40-µm cell strainer, followed by analysis using a flow cytometer (Cyto-FLEX with the software (CytExpert v2.4), Beckman Coulter) with a 488-nm laser for excitation. Data were collected from the fluorescein isothiocyanate (FITC) detector (for the oxidized form of BODIPY) with a 525/40 nm bandpass filter, or from the phycoerythrin (PE) detector (for the reduced form of BODIPY) with a 585/42 nm bandpass filter. At least 10,000 events were analyzed per sample. Data were analyzed using FlowJo Software. The ratio of fluorescence of C11-BODIPY 581/591 (lipid peroxidation) (FITC/PE ratio (oxidized/reduced)) was calculated using the median value of each channel[30].

### Live-cell imaging

Pfa1 cells (5,000–10,000 cells) were seeded on a µ-Dish 35 mm, low (ibidi, cat. no. 80136), and incubated overnight. The next day, the cell culture medium was changed to fresh medium. Live-cell imaging was performed using 3D Cell Explorer (Nanolive) with the software Evev1.8.2. During imaging, the cells were maintained at 37 °C and a 5% $CO_2$ atmosphere using a temperature-controlled incubation chamber. After recording one image, a 100-fold concentration of viFSP1 in DMEM

was added to the dishes (the final concentration was 10 μM viFSP1), and then recording was continued. For ferroptosis suppression, cells were pretreated 0.5 μM Lip-1 for 15 min before recording. Images were taken every 5 min for more than 6 h.

## Cell lysis and immunoblotting

Cells were lysed in LCW lysis buffer (0.5% Triton X-100, 0.5% sodium deoxycholate salt, 150 mM NaCl, 20 mM Tris-HCl, 10 mM EDTA, 30 mM Na-pyrophosphate tetrabasic decahydrate), supplemented with protease and phosphatase inhibitor cocktail (cOmplete and phoSTOP, Roche, cat. no. 04693116001 and cat. no. 4906837001), and centrifuged at 20,000g, 4 °C, for 30 min to 1 h. The cell lysate was sampled by dissolved with 6×SDS loading buffer (375 mM Tris-HCl, 9% SDS, 50% glycerol, 9% β-mercaptoethanol, 0.03% bromophenol blue, pH 6.8). After boiling at 95 °C for 3 min, the samples were resolved on 12% SDS–PAGE gels (Bio-Rad, cat. no. 4568043 or cat. no. 4568046) and subsequently electroblotted onto a polyvinylidene difluoride (PVDF) membrane (Bio-Rad, cat. no. 1704156 or cat. no. 1704274), following the manufacturer's protocol. The membranes were incubated in the blocking buffer, 5% milk (Roth, cat. no. T145.2) in TBS-T (20 mM Tris-HCl, 150 mM NaCl, and 0.1% Tween-20), then probed with the primary antibodies. The primary antibodies were diluted in antibody-dilution buffer (5% BSA, 0.1% NaN₃ (Sigma, cat. no. S2002) in TBS-T) and were against GPX4 (1:1000, Abcam, cat. no. ab125066), valosin containing protein (VCP, 1:10,000, Abcam, cat. no. ab109240), HA tag (1:1,000, rat IgG1, clone 3F10, developed in-house), human FSP1 (1:1,000, Santa Cruz, cat. no. sc-377120, AMID) or human FSP1 (1:5, rat IgG2a, clone AIFM2 6D8, developed in-house), human and mouse FSP1 (1:5, rat IgG2a, clone AIFM2 1A1-1, developed in-house) and human and mouse FSP1 (1:5, rat IgG2b, clone AIFM2 14D7, developed in-house), or were diluted in 5% milk in TBS-T against horseradish-peroxidase-conjugated β-actin (1:50,000, Sigma, cat. no. A3854) overnight. After the membrane was washed once, it was probed with secondary antibodies (1:1,000–1:5,000, Cell Signaling, cat. no. 7074S for rabbit; cat. no.7076S for mouse, and 1:1,000 for anti-rat IgG1b and 2a/b, developed in-house) diluted in 5% skim milk in TBS-T. The antibody–antigen complexes were detected by the ChemiDoc Imaging System with Image Lab v6.0 (Bio-Rad). Representative images are shown after being adjusted to the appropriate brightness and angle using ImageJ/Fiji software (ver.1.53).

## Construction of expression plasmids

All plasmids in this study were constructed using standard molecular biology techniques, and were verified by sequencing : human *AIFM2* cDNA (NM_001198696.2, C>T:1008) and codon-optimized *Mus musculus* (mouse) FSP1 (NP_001034283.1), *Rattus norvegicus* (rat) FSP1 (NP_001132955.1) and *Gallus gallus* (chicken) FSP1 (XP_421597.1) were cloned or synthesized into p442-IRES-blast vectors[8,17]. For generating mutants or subcloning, DNA was first amplified by KOD One (Sigma, cat. no. KMM-201NV), and PCR products were purified by Wizard SV Gel&PCR Clean-up System (Promega, cat. no.A9285). Ligation reactions of PCR products with digested vectors were performed using In-Fusion cloning enzymes (Takara Bio, cat. no.639649 or 638948). Subsequently, reaction mixtures were transformed into NEB stable competent cells (NEB, cat. no.C3040H). Plasmids were isolated using the QIAprep Spin Miniprep Kit (QIAGEN, cat. no.27106), followed by sequencing.

## Lentiviral production and transduction

HEK293T cells were used to produce lentiviral particles. The ecotropic envelope protein of the murine leukemia virus (MLV) was used for murine-derived cells. A third-generation lentiviral packaging system consisting of transfer plasmids, envelope plasmids (pEcoEnv-IRES-puro or pHCMV-EcoEnv (Addgene, cat. no.15802) (ecotropic particles)) and packaging plasmids ((pMDLg_pRRE and pRSV_Rev, or psPAX2 (Addgene, cat. no.12260)) were co-lipofected into HEK293T cells using transfection reagents (PEI MAX (Polysciences, cat. no. 24765)

or X-tremeGENE HP agent (Roche, cat. no. 06366236001)). Cell culture supernatants containing viral particles were collected at 2–3 d post-transfection and filtered through a 0.45-μm PVDF filter (Millipore, cat. no. SLHV033RS), and were then stored at −80 °C.

Cells were seeded on 12- or 6-well plates with lentivirus supplemented with 10 μg mL⁻¹ protamine sulfate overnight. On the next day, the cell-culture medium was replaced with fresh medium containing appropriate antibiotics, such as puromycin (Gibco, cat. no. A11138-03; 1 μg mL⁻¹) and blasticidin (Invitrogen, cat. no. A1113903; 10 μg mL⁻¹), and transduced cells were selected until non-transduced cells were completely dead.

## Mutational screens

Mutated *AIFM2* cDNA (NM_001198696.2, C>T:1008) with the adapter sequence for p442 was amplified from the virus expression vector p442-hFSP1-IRES-Blast[8] by error-prone PCR using the GeneMorph II Random Mutagenesis Kits (Agilent cat. no. 200550). The optimal mutagenesis rate (3–5 DNA mutations per gene) was achieved using PCR under the following conditions: (1) 95 °C for 120 s, (2) 95 °C 30 s, (3) 60 °C 30 s, (4) 72 °C for 75 s, (5) 72 °C 10 min; the cycles in steps 2–4 were repeated 25 times, and all other steps were performed once. After the isolation of PCR products using agarose electrophoresis and clean-up using the Wizard SV Gel&PCR Clean-up System, the FSP1 fragment was ligated into a digested p442-IRES-blast vector using In-Fusion enzyme (Takara, cat. no. 639649) under the following conditions: 400 ng DNA insert, 200 ng vector and 8 μL enzyme in 40 μL reaction volume, at 50 °C for 15 min. Then, the insert–vector mixture was transformed into NEB stable cells, which were incubated at 37 °C for 1 h. After the induction of antibiotic-resistant genes, cells were applied to 8 plates (Thermo Fisher, cat. no. 240835) and cultured with ampicillin selection growth medium at 30 °C overnight. On the next day, colonies were collected, and DNA was isolated using the QIAGEN Plasmid Maxi Kit (Qiagen, cat. no. 12163).

As described above, a third-generation lentiviral packaging system consisting of FSP1 mutant library plasmids, envelope plasmids (pEcoEnv-IRES-puro) and packaging plasmids (pMDLg_pRRE and pRSV_Rev) were co-lipofected into HEK293T cells using transfection reagents (X-tremeGENE HP agent). Cell culture supernatants containing viral particles were collected at 2 d post-transfection and filtered through a 0.45-μm PVDF filter, and were then stored at −80 °C.

Pfa1 cells were seeded into ten T-175 flasks ($1.0 \times 10^6$ cells per flask, $2.0 \times 10^7$ cells in total) with medium containing 10 μg mL⁻¹ protamine and lentivirus containing FSP1 mutants and were transduced with extremely low infection efficiency (MOI = approximately 0.1, calculated as previously described[43]). The next day, the medium was replaced with the fresh medium containing blasticidin (12.5 μg mL⁻¹) and puromycin (1 μg mL⁻¹). After selection using blasticidin for 3 d, 10 million cells were collected as a control group; at the same time, 20 million cells were seeded into five flasks ($2.0 \times 10^6$ cells per flask, $2.0 \times 10^7$ cells in total) with 500 nM RSL3. After the induction of ferroptosis by RSL3 for 2 d, 10 million cells were collected as the RSL3-treated group. Then, 10 million cells per condition were treated with individual FSP1 inhibitors (5 μM for the first 5 d and 10 μM thereafter) and with 500 nM RSL3 and 1 μM 4-OH Tam so that the Pfa1 cells became fully resistant to FSP1 inhibitors while still expressing functional FSP1. Then, surviving Pfa1 *Gpx4*-KO cells overexpressing resistant FSP1 mutants were expanded, and 10 million cells per condition were collected and stored at −80 °C.

After collecting cells from all conditions, cells were lysed in lysis buffer (50 mM Tris, 50 mM EDTA, 1% SDS, pH 8) with proteinase K (100 μg mL⁻¹) at 55 °C overnight. On the next day, RNase A (50 μg mL⁻¹) was added and incubated for 30 min at 37 °C to digest RNA. Then, the equivalent volume of phenol:chloroform:isoamyl alcohol (25:24:1) (Roth, cat. no. A156.2) was added. The solution was briefly vortexed and centrifuged for 10 min at 16,000g to separate DNA from RNA and proteins. The top phase of the DNA-containing solution was carefully collected in the new tubes and DNA was precipitated by the 2× volumes

of 75 mM NaCl in ethanol by centrifugation for 10 min at 16,000$g$. The pellet was then washed with 70% ethanol, followed by centrifugation. After drying the remaining ethanol, the pellet was dissolved in 200 µL TE buffer per condition and incubated at 65 ℃ for 1 h. Finally, the DNA of the *AIFM2* region (approximately 1,500 base pairs) was amplified by KOD One (Sigma, Cat. no. KMM-201NV) and purified as described above.

NGS library preparation was performed using ThruPLEX DNA-Seq HV PLUS kit (Takara, cat. no. R400782) with minor optimization. After the preparation of the library, the DNA was purified using NucleoMag NGS Clean-up and Size Select (Th. Geyer, cat. no. 11833159) for subsequent NGS. NGS was performed by the core facility in Helmholtz Munich.

### NGS data analysis of human *AIFM2* cDNA
Paired-end sequencing was performed in different conditions using an Illumina NovaSeq 6000 instrument using ThruPLEX DNA-Seq HV PLUS kit (see Supplementary Table 1). The program FastQC (v0.11.7) (http://www.bioinformatics.babraham.ac.uk/projects/fastqc) was applied to the resulting FASTQ files to identify sequences that were over-represented (Illumina adapter) and to exclude them from further analysis. We used the Trimmomatic V.039 tool[44] with the following options (ILLUMINACLIP:TruSeq2-PE_extended.fa:2:30:10 LEADING:3 TRAILING:3 SLIDINGWINDOW:4:15 MINLEN:36) to trim paired-end data (see Supplementary Table 1). Trimmed paired-end reads (original length, 151 bp) were aligned to the reference *AIFM2* cDNA sequence of 1,210 bp with the Burrows-Wheeler Alignment Tool (BWA), version 0.7.17-r1188 (ref. 45). First, an index was generated for the reference sequence using the command 'bwa index'. Then, by applying the subcommand 'mem', BWA outputs the final alignment in the SAM (Sequence Alignment/Map) format. Aligned reads were converted to the BAM (Binary Alignment Map) format and sorted by leftmost chromosomal coordinates with the program SAMtools version 1.2 (ref. 46) using the commands 'samtools view' and 'samtools sort', respectively. From sorted BAM files, the coverage information per base for the FSP1 reference sequence was extracted by applying IGVtools of the Integrative Genomics Viewer (IGV), version 2.11.9 (ref. 47). The command 'igvtools count' with the options (-w 1 and–bases) was used to produce an output file in the WIG (wiggle) format for each sorted BAM file. The mutation frequency at each position (Xi) of the FSP1 reference sequence was calculated from the sum of mutated nucleotides at the corresponding position of the reference sequence in the alignment divided by the sum of all nucleotides (A,C,G,T,N) at this location. The count information per base stored in the wiggle files served as input for a custom written R script for the calculation of the mutation frequency. The tab-delimited output file of the R script contains the number of each nucleotide and deletions and insertions, as well as the sum of all nucleotides and the mutation frequency from the alignment (columns) at each position of the FSP1 sequence (rows).

$$\text{Mutation frequency } (Xi) = N_{\text{mut}}(Xi)/N_{\text{all}}(Xi)$$

Where Xi is the position of *AIFM2*, $N_{\text{mut}}$ the sum of mutated nucleotides at Xi and $N_{\text{all}}$ is the sum of all nucleotides at Xi. The $Z$ score was calculated using mean ($\mu$) and standard deviation ($\sigma$) as follows:

$$Z = [Xi - \mu_{\text{RSL3 or iFSP1s}}]/\sigma_{\text{RSL3 or iFSP1s}}$$

$$\mu_{\text{RSL3}} = \text{Mean(Mutationfrequency}_{\text{RSL3}}/\text{Mutationfrequency}_{\text{ctrl}})$$

$$\mu_{\text{iFSP1s}} = \text{Mean (Mutationfrequency}_{\text{iFSP1s}}/\text{Mutationfrequency}_{\text{RSL3}})$$

$\sigma_{\text{RSL3}}$
$$= \text{Standard Deviation (Mutationfrequency}_{\text{RSL3}}/\text{Mutationfrequency}_{\text{ctrl}})$$

$\sigma_{\text{iFSP1s}}$
$$= \text{Standard Deviation (Mutationfrequency}_{\text{iFSP1s}}/\text{Mutationfrequency}_{\text{RSL3}})$$

Where iFSP1s represents iFSP1 or viFSP1, and ctrl is the control. Then, amino acid substitutions were investigated in light of the possible DNA alternation using the codon table (Supplementary Table 1).

### Screening of mouse FSP1 inhibitors
Pfa1 and Pfa1 *Gpx4*-KO cells stably overexpressing murine FSP1 were seeded on separate 384-well plates (500 cells per well), and screened with a library of small-molecule inhibitor compounds, as reported previously[8]. Viability of the different cell lines was assessed 48 h after treatment using AquaBluer. Compounds showing selective lethality in Pfa1 *Gpx4*-KO cells stably overexpressing murine FSP1 were then validated in cell viability and in vitro FSP1 enzymatic assays, as described above.

### CRISPR–Cas9-mediated gene knockout
Single guide RNAs (sgRNA) vectors for human GPX4, mouse Gpx4 and mouse Fsp1 were established, and KO cells were generated as previously reported[10,17]. SW620 *GPX4*-KO cells were transiently co-transfected with the desired sgRNAs expressing lentiCRISPRv2-blast or lentiCRISPRv2-puro using the X-tremeGENE HP. One day after transfection, cells were selected with puromycin (1 µg mL$^{-1}$) and blasticidin (10 µg mL$^{-1}$) until non-transfected cells were dead. Single-cell clones were isolated in serial dilutions, and KO clones were confirmed by immunoblotting.

### Stable expression of genes by transfection
4T1 *Gpx4*-KO cells and B16F10 *Gpx4-KO Fsp1*-KO (DKO) cells were transfected with 141-IRES-puro, 141-hFSP1-IRES-puro, 141-mFsp1-IRES-puro, 141-mGpx4-IRES-puro or 141-mGpx4 U46C-IRES-puro vectors using the X-tremeGENE HP agent. One day after transfection, cells were selected and cultured in the presence of puromycin (1 µg mL$^{-1}$) and absence of Lip-1 to select for stable FSP1- or GPX4-expressing cells.

### In silico modeling and structural analysis
A predicted human FSP1 structure was obtained from the AlphaFold2 database (https://alphafold.ebi.ac.uk)[28]. To yield the superimposed structure of FSP1 with its cofactor flavin adenine dinucleotide (FAD), NADH and ubiquinone, the structure of the FSP1 ortholog NDH-2 (PDB: 4G73 ref. 29 or 5NA1 ref. 48) was aligned to FSP1 using Pymol v2.5.2 (Schrödinger), and the positions of FAD, NADH and ubiquinone were extracted and embedded into FSP1 structure[19] or another NDH-2 (PDB: 4NWZ ref. 49). The in silico modeling for iFSP1 and viFSP1 were conducted by modeling software SeeSAR v12.1 (BioSoveIT)[19]. After visual inspection, the most viable poses were selected, and then docking molecules were exported as PDB files. Depictions of all docking or superimposed structures were made using Pymol.

### Protein alignment of FSP1 orthologues
The human FSP1 sequence and its orthologs were obtained from UniProt (https://www.uniprot.org), NCBI (https://www.ncbi.nlm.nih.gov/gene/) and the PDB (https://www.rcsb.org), then aligned and visualized using JalView[50] (v2.11.2.6).

### Genome DNA extraction and sequencing
Genomic DNA from SKOV3 was extracted using DNAzol (Fisher Scientific, cat. no.15413379), according to the manufacturer's instructions. Sequencing was performed by Eurofins genomics.

### Statistics and reproducibility
All data shown are the mean ± s.e.m. or mean ± s.d., and the numbers (*n*) in each figure legend represents biological replicates or technical replicates. All experiments (except for the mutational analysis) were performed independently at least twice. One-way or two-way ANOVA followed by Dunnett's or Tukey's multiple-comparisons test were performed using GraphPad Prism 9 (GraphPad Prism) (see the figure legends for more details). The results of the statistical analyses are represented in each figure. *P* < 0.05 was considered statistically

significant. No statistical method was used to predetermine sample size. No data were excluded from the analyses. The experiments were not randomized. The investigators were not blinded to allocation during experiments or outcome assessment.

## Reporting summary
Further information on research design is available in the Nature Portfolio Reporting Summary linked to this article.

## Data availability
All data are available in the article and the supplementary information, and from the corresponding author on reasonable request. Gel source images are shown in Supplementary Figure 2. Human cancer cell line data were mined from the DepMap (https://depmap.org/portal/) or COSMIC (https://cancer.sanger.ac.uk/cosmic) databases. Murine cancer cell line data was mined from the TISMO database (http://tismo.cistrome.org)[51]. The sequence data from this study have been submitted to NCBI BioProject (https://www.ncbi.nlm.nih.gov/bioproject) under BioProject ID PRJNA942499. Source data are provided with this paper.

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

## Acknowledgements

We are grateful to all current and former members of the Conrad Laboratory for providing valuable materials and fruitful discussions. This work was supported by Deutsche Forschungsgemeinschaft (DFG) (CO 291/7-1 and the Priority Program SPP 2306 (CO 291/9-1, no. 461385412; CO 291/10-1, no. 461507177)) and PR 1752/3-1 to B.P., the German Federal Ministry of Education and Research (BMBF) FERROPATH (01EJ2205B), and the European Research Council (ERC) under the European Union's Horizon 2020 research and innovation program (grant agreement no. GA 884754) to M.C. MC38 cells were kindly gifted from P. Agostinis (KU Leuven, Belgium). Rat1 cells were kindly gifted from Medizinische Hochschule Hannover. Huh7 cells were kindly gifted from R. Schneider (Helmholtz Munich).

## Author contributions

T.N., M.S., B.P. and M.C. conceived the study and wrote the manuscript. T.N., E.M., N.Y., J.W. and E.L. performed in vitro and cell experiments. A.S.D.M. expressed FSP1 in cells and purified recombinant FSP1. T.N. and P.S. performed in silico analysis. T.N., S.D. and D.T. designed and performed mutational screens and analyzed the resulting data. T.N.X.d.S. and J.P.F.A made murine FSP1 expression cells for compound screening. All authors read and agreed on the content of the paper.

## Funding

## Competing interests
M.C., B.P. and P.S. are co-founders and shareholders of ROSCUE Therapeutics. The other authors declare no competing interests.

## Additional information
**Extended data** is available for this paper at https://doi.org/10.1038/s41594-023-01136-y.

**Correspondence and requests for materials** should be addressed to Marcus Conrad.

**a**

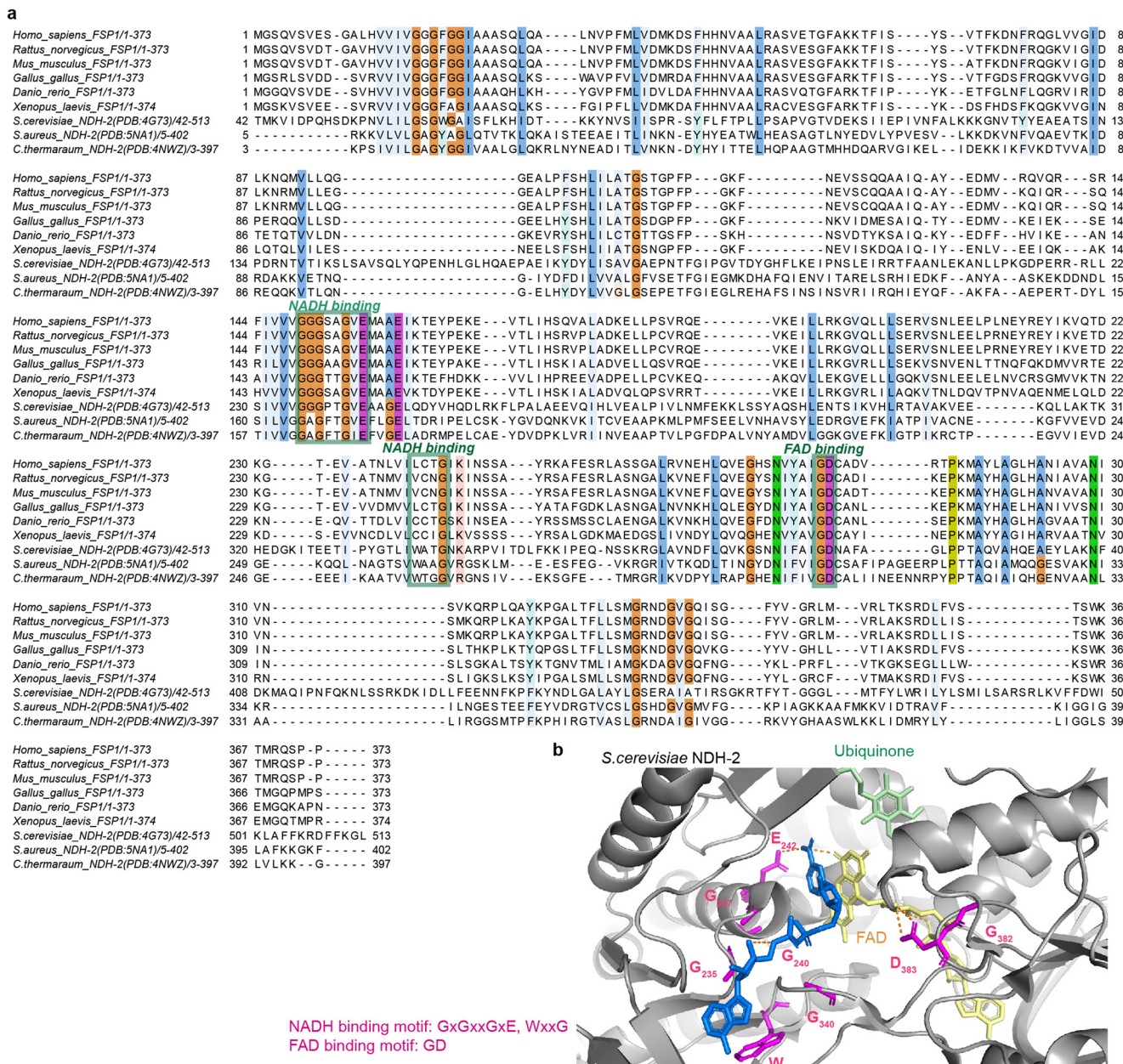

**Extended Data Fig. 1 | Multiple protein alignment among different FSP1 orthologues.** a. Protein alignment represents the difference among FSP1 orthologues: *Homo sapiens* (Human), *Mus musculus* (Mouse), *Rattus norvegicus* (Rat), *Gallus gallus* (Chicken), *Xenopus laevis* (Frog), *Danio rerio* (zebrafish), *Saccharomyces cerevisiae* (*S. cerevisiae*) Ndi1, *Caldalkalibacillus thermarum* (*C. thermarum*) NDH-2, and *Staphylococcus aureus* (*S. aureus*) NDH-2. Highly conserved amino acids are highlighted in different colors. The NADH binding

(GxGxxGxE, WxxG) and FAD binding (GD) consensus sequences (green boxes) are highly conserved among the different orthologues. b. Crystal structure of *S. cerevisiae* Ndi1 (PDB: 4G73) with consensus NADH and FAD binding motif. The cofactors, flavin adenine dinucleotide (FAD, yellow), nicotinamide adenine dinucleotide (NADH, blue) and ubiquinone (CoQ$_5$, green) are highlighted. The expected hydrogen bond was generated by Pymol.

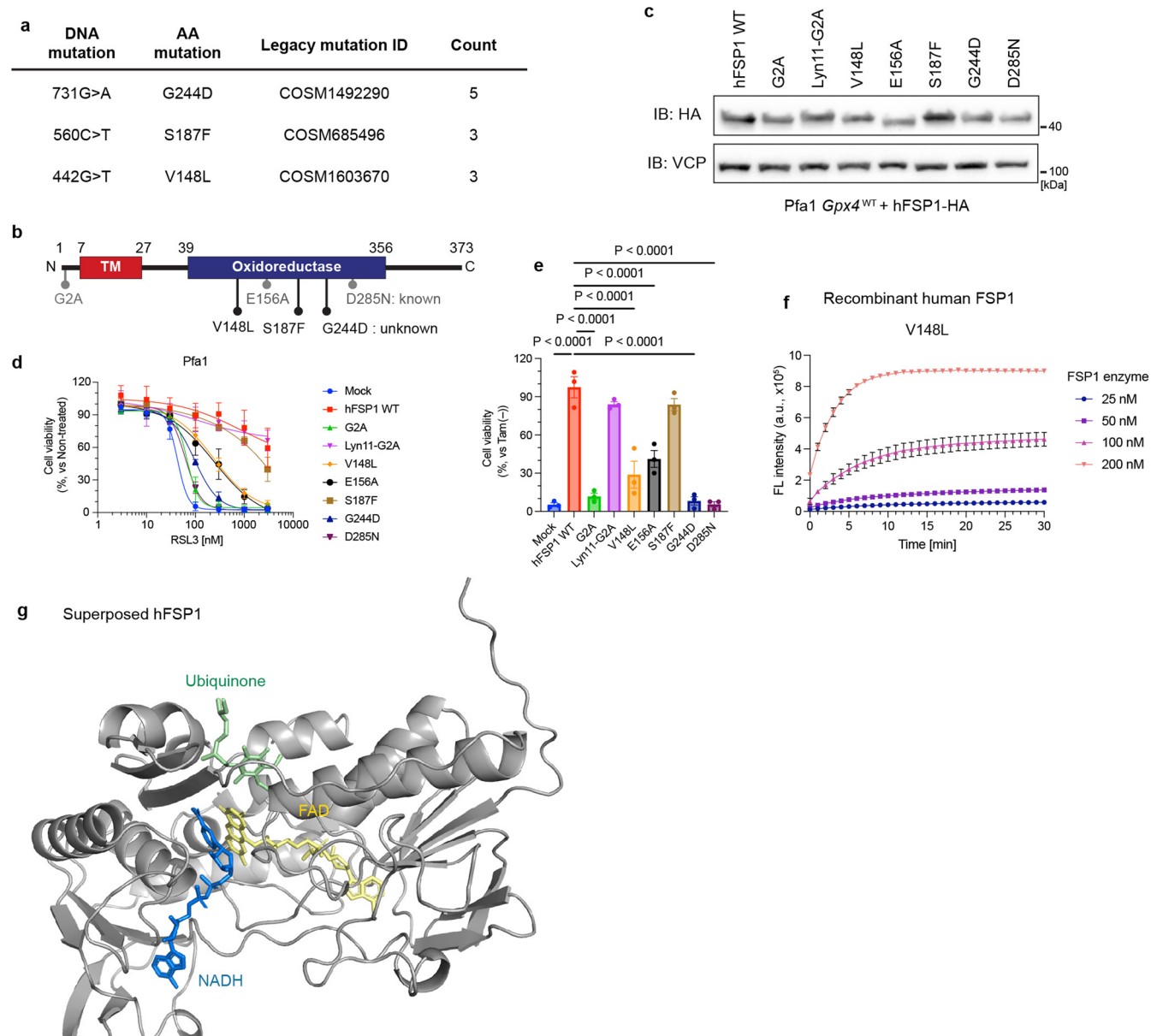

**Extended Data Fig. 2 | Analysis of somatic FSP1 mutations reported in cancer patients.** a. Somatic mutations of *AIFM2* found in cancer patients using the COSMIC database. b. Schematic FSP1 enzyme representation with annotated functional mutations as reported. c. Immunoblot analysis of FSP1 and VCP expression in Pfa1 cells stably overexpressing wildtype hFSP1 or its mutant variants from a single experiment. d. Cell viability in Pfa1 cells stably overexpressing wildtype hFSP1 or its mutant variants treated with RSL3 for 24 h. e. Cell viability was measured after treating Pfa1 cells stably overexpressing wildtype hFSP1 or its mutant variants with or without 1 µM Tam for 72 h. Data was normalized by each group of non-treatment with Tam. Data represents the mean ± SEM of 3 wells of a 96-well plate from one out of 4 (d) or 3 (e) independent experiments. p values were calculated from one-way ANOVA followed by Dunnett's multiple comparison test. f. Reduction of resazurin in the presence of FSP1 V148L at the indicated concentrations. Data represents the mean ± SD of 3 wells of a 96-well plate from one out of 3 independent experiments. g. Superimposed human FSP1 structure from AlphaFold2 database (https://alphafold.ebi.ac.uk). The cofactors, flavin adenine dinucleotide (FAD, yellow), nicotinamide adenine dinucleotide (NADH, blue) and ubiquinone (CoQ$_5$, green) were embedded from the structure of the yeast orthologue, NDH-2 (Ndi1) (PDB: 4G73).

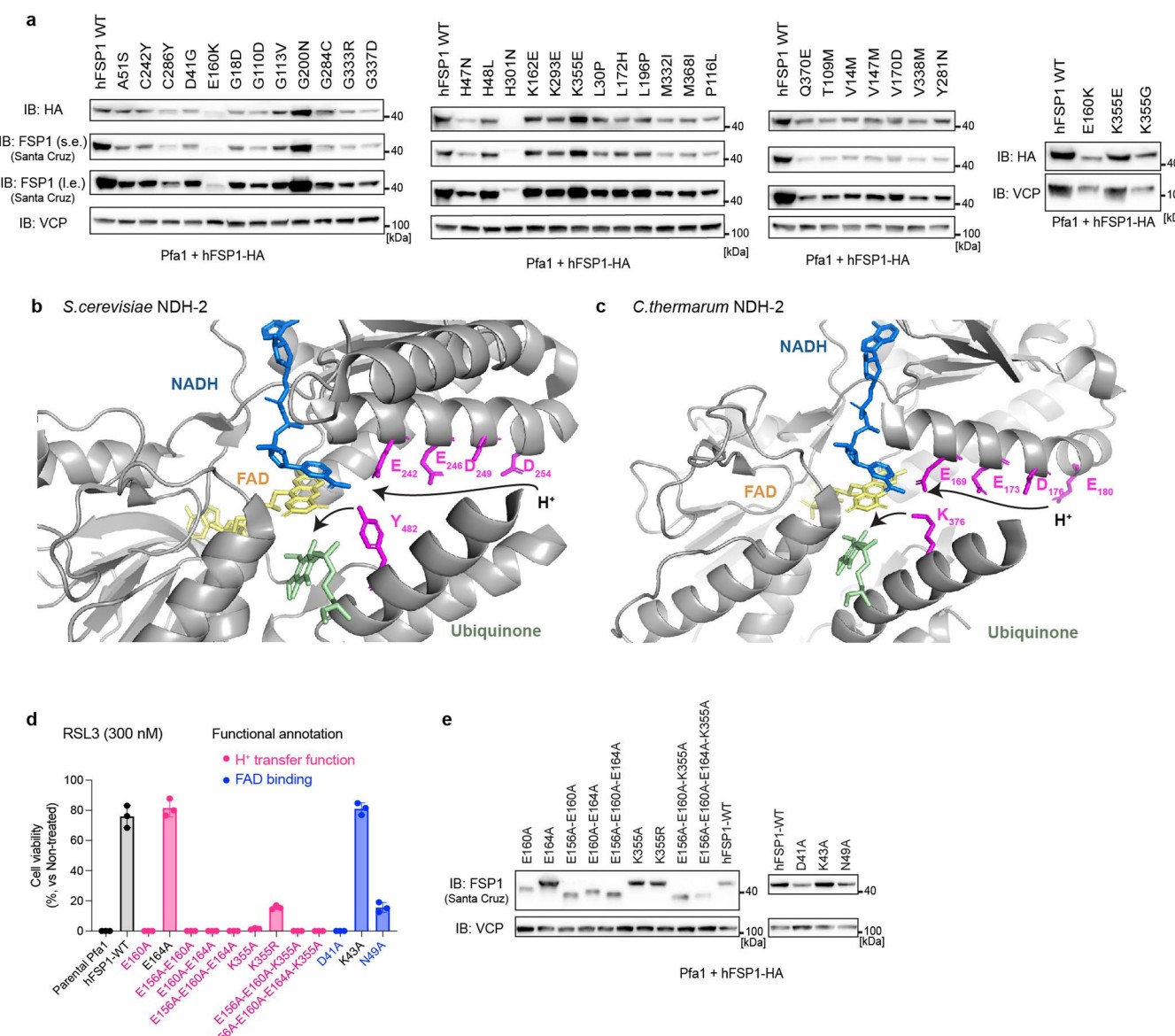

**Extended Data Fig. 3 | Dysfunctional FSP1 mutations and proposed proton transfer function in NDH-2s.** a. Immunoblot analysis of FSP1 (using AMID (Santa Cruz) and HA antibodies) and VCP expression in Pfa1 cells stably overexpressing wildtype hFSP1 or its mutant variants. For immunoblot of FSP1, s.e. and l.e. represent short and long exposure, respectively. b. Crystal structure of *Saccharomyces cerevisiae* (*S. cerevisiae*) NDH-2 (PDB: 4G73) with its cofactors, FAD (yellow), NADH (blue), and CoQ₅ (green). The proton transfer via sequential carboxylic residues in the α-helix (that is, E242/E246/D249/D294) and subsequent final protonation from Y482 to ubiquinone are indicated with black arrows. c. Crystal structure of *Caldalkalibacillus thermarum* (*C. thermarum*)

NDH-2 (PDB: 4NWZ) with its cofactors, FAD (yellow), NADH (blue), and CoQ₅ (green). The proton transfer via sequential carboxylic residues in the α-helix (that is, E169/E173/D176/E180) and subsequent final protonation from K376 to ubiquinone are indicated with black arrows. d. Cell viability was measured after treating Pfa1 cells stably overexpressing wildtype hFSP1 or its mutant variants with or 300 nM RSL3 for 24 h. Data was normalized by each group of non-treatment with RSL3. Data represents the mean ± SD of 3 wells of a 96-well plate from one out of 3 independent experiments. e. Immunoblot analysis of FSP1 (Santa Cruz) and VCP expression in Pfa1 cells stably overexpressing wildtype hFSP1 or its mutant variants in cells. Data is shown from a single experiment (a,e).

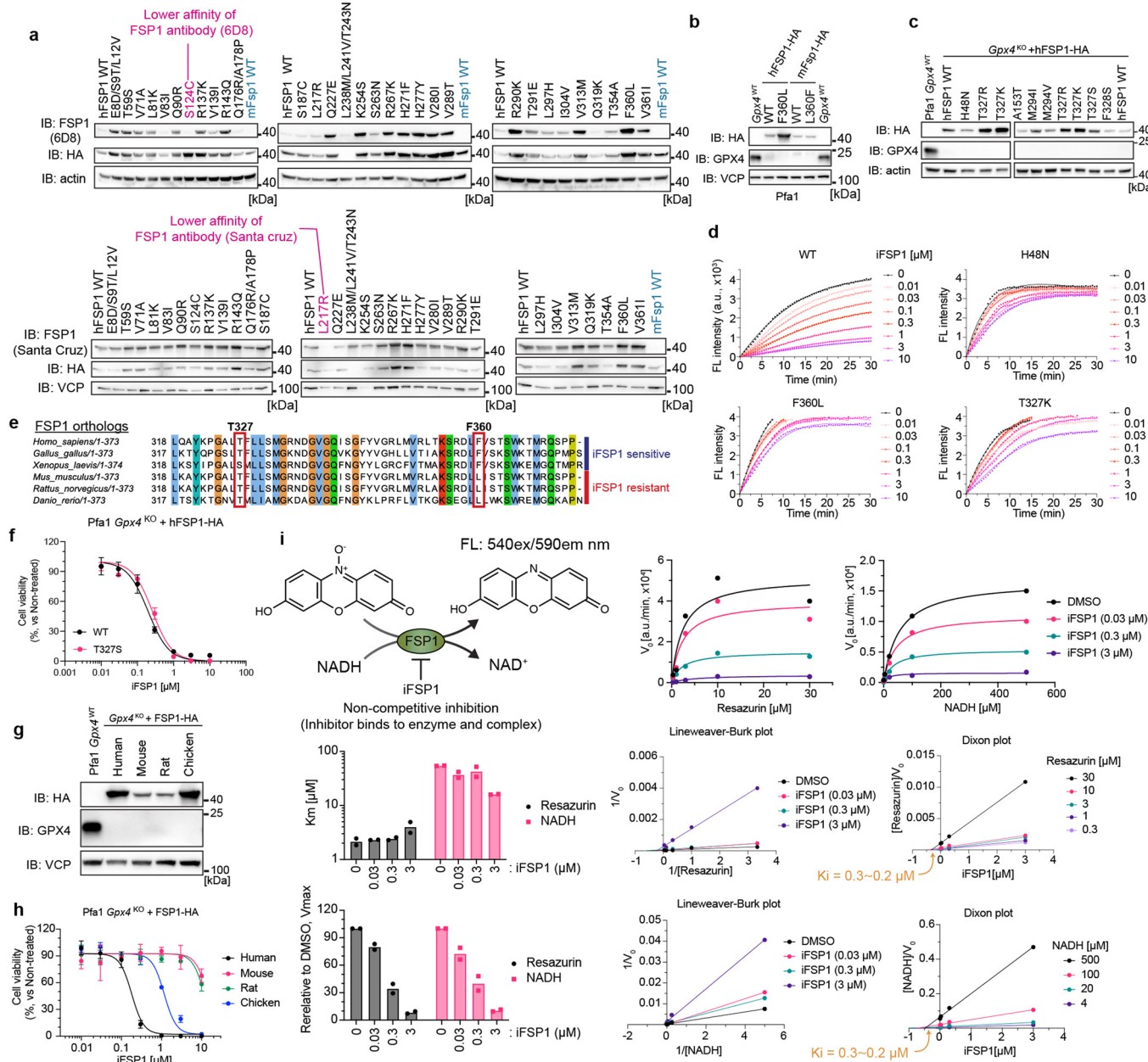

**Extended Data Fig. 4 | Identification of the iFSP1 binding pocket via site-directed mutagenesis.** a. Immunoblot analysis of FSP1 (clone 6D8, AMID: Santa Cruz, and HA), actin, and VCP expression in Pfa1 cells stably overexpressing wildtype hFSP1 or its mutant variants. Each human specific FSP1 antibody (6D8 and Santa Cruz) showed a lower affinity against the highlighted human FSP1 mutants similar to mouse wildtype FSP1. b. Immunoblot analysis of GPX4, FSP1 (HA), and VCP expression in Pfa1 cells stably overexpressing wildtype hFSP1 and its mutant variants. c. Immunoblot analysis of GPX4, FSP1 (HA), and actin expression in Pfa1 cells stably overexpressing wildtype hFSP1 and its mutant variants. Data is shown from a single experiment (a-c). d. Representative resazurin reduction assay in the presence of wildtype FSP1 or its mutant variants with the indicated concentrations of iFSP1. Data represents a single well of a 96-well plate from one out of 3 independent experiments. e. Alignment of different FSP1 orthologues. Note, T327 and F360 (in human position) are different among species. f. Cell viability of Pfa1 Gpx4-KO cells stably expressing wildtype hFSP1

or the T327 mutant treated with iFSP1 for 24 h. Data represents the mean ± SD of 3 wells of a 96-well plate from one out of 2 independent experiments. g. Immunoblot analysis of GPX4, FSP1 (HA), and VCP expression in Pfa1 cells stably overexpressing wildtype FSP1 orthologues from a single experiment. h. Cell viability of Pfa1 Gpx4-KO cells stably expressing the different wildtype FSP1 orthologues treated with iFSP1 for 24 h. Data represents the mean ± SD of 3 wells of a 96-well plate from one out of 3 independent experiments. i. Enzyme kinetics in vitro assay. Schematic representation of the FSP1 enzyme activity assay using resazurin and NADH as the substrates. Representative reduction kinetics of resazurin, bar plots of the $K_m$ and relative $V_{max}$ values, Lineweaver-Burk plot, and Dixon plot in the presence of different concentrations of either NADH or resazurin with iFSP1 at indicated concentrations. Reduction kinetics, Lineweaver-Burk plot, and Dixon plot represent a single well of a 96-well plate from one out of 2 independent experiments. Bar plots represent the mean from 2 independent experiments.

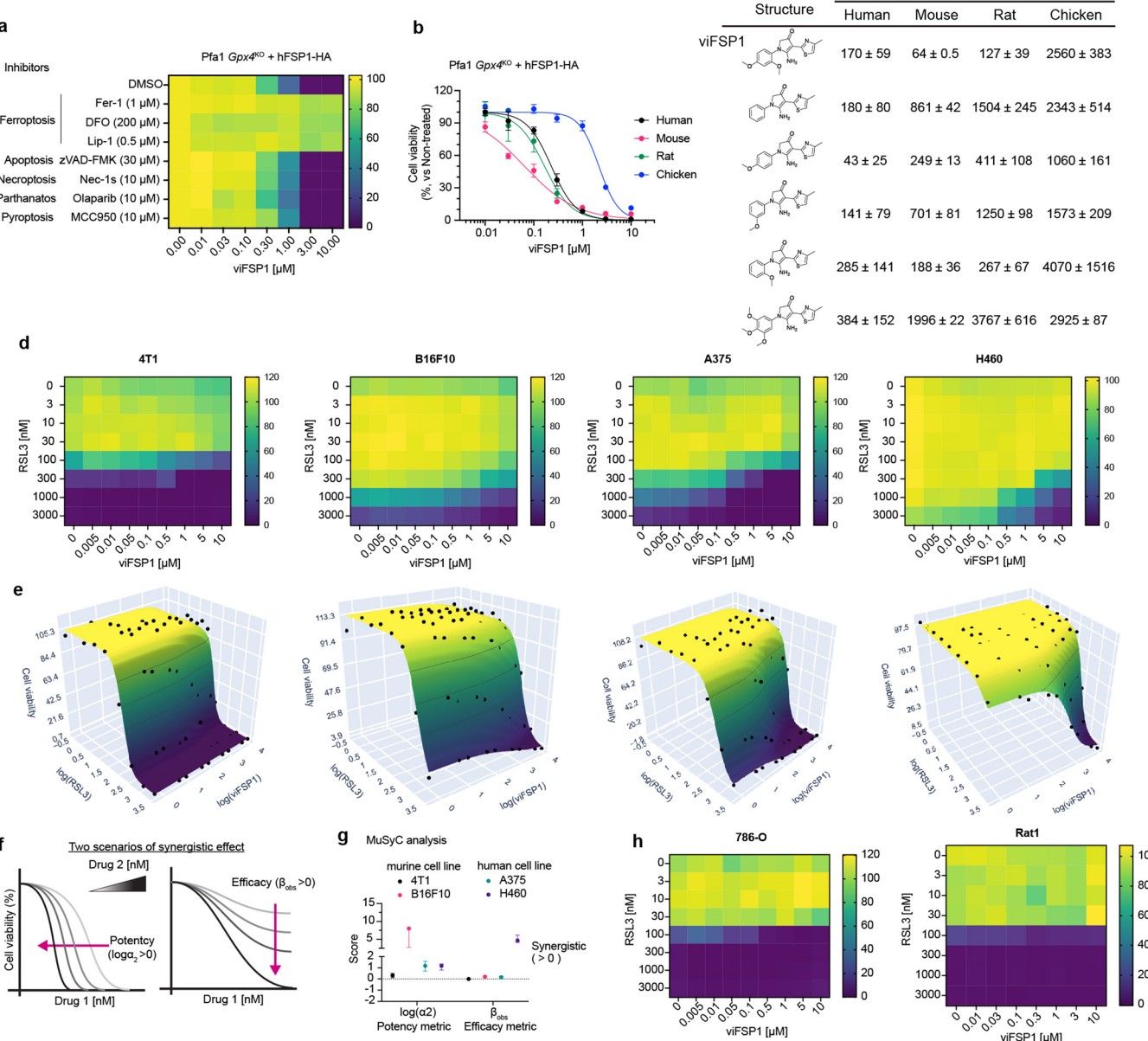

**Extended Data Fig. 5 | viFSP1 is a potent FSP1 inhibitor in different cell lines.**
a. Cell viability of Pfa1 *Gpx4*-KO cells stably overexpressing hFSP1 mutants treated with viFSP1 for 48 h with indicated cell death inhibitors. Ferrostatin-1 (Fer-1), deferoxamine (DFO) and liproxstatin-1 (Lip-1) were used as ferroptosis inhibitors. b. Cell viability of Pfa1 *Gpx4*-KO cells stably overexpressing wildtype FSP1 enzyme from different organisms treated with viFSP1 for 48 h. Data represents the mean ± SD of 3 wells of a 96-well plate from one out of 2 independent experiments (See also Supplementary Videos 1-8). c. An initial structure–activity relationship study (SAR) for viFSP1. Pfa1 *Gpx4*-KO cells stably expressing wildtype FSP1 from different species were treated with the different viFSP1 analogs for 48 h. EC₅₀ values were obtained from 3 wells of a 96-well plate

from a single experiment and calculated by GraphPad Prism9. d. Heat maps represent cell viability of 4T1, B16F10, A375, and H460 cells treated with viFSP1 and RSL3 for 48 h. Data represents the mean of 4 wells of a 384-well plate from 3 independent experiments. e. 3D-Heat maps represent the same data as d and are visualized by MuSyC (https://musyc.lolab.xyz)[38,39]. f. Schematic models for the synergistic effect of 2 given compounds. g. Synergistic potency and efficacy. Data is calculated by MuSyC using the data from d. Data represent the mean ± 95% confidence intervals of three independent experiments. h. Heat map represents cell viability of 786-O and Rat1 cells treated with viFSP1 and RSL3 for 48 h. Data represents the mean of 4 wells of a 384-well plate (786-O) or a single well of a 96-well plate (Rat1) from one out of 3 or 2 independent experiments.

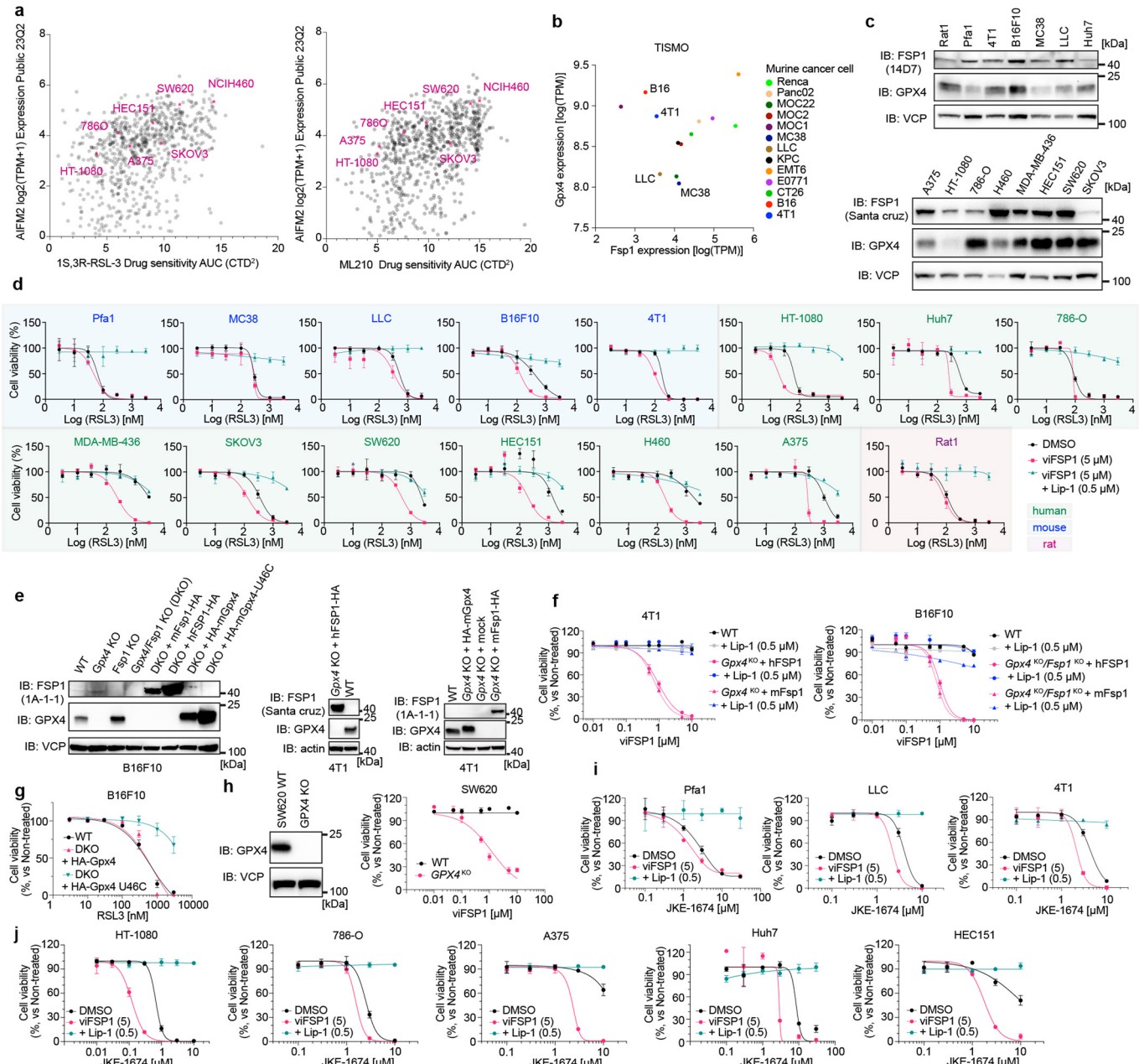

**Extended Data Fig. 6 | viFSP1 shows the synergistic effect toward ferroptosis in numerous cancer cells.** a. DepMap database analysis. *AIFM2* (*FSP1*) expression and sensitivity of GPX4 inhibitors (RSL3 and ML210) are shown and cell lines used in this study are highlighted. b. TISMO data analysis. The baseline expression of *Gpx4* and *Fsp1* are shown and cell lines used in this study are highlighted. c. Immunoblot analysis of GPX4, FSP1, and VCP expression in various cell lines (from a single experiment). d. Cell viability of various cell lines treated with RSL3 and/or viFSP1 for 24 h. Lip-1 was used as a ferroptosis inhibitor. e. Immunoblot analysis of GPX4, FSP1, actin and VCP expression in B16F10 and 4T1 cells from a single experiment. DKO: *Gpx4*-KO/*Fsp1*-KO. f. Cell viability of 4T1 *Gpx4*-KO cells stably expressing hFSP1 or mFSP1, and B16F10 DKO cells stably expressing hFSP1 or mFSP1 treated with viFSP1 for 48 h. Lip-1 was used as ferroptosis inhibitor. g. Cell viability of B16F10 DKO cells stably expressing mGPX4 or mGPX4 U46C treated with RSL3 for 24 h. h. Representative immunoblot analysis of GPX4, and VCP expression in SW620 cells from a single experiment (left). Cell viability of SW620 wildtype cells or *GPX4*-KO treated with viFSP1 for 48 h (right). i. Cell viability of murine cell lines treated with JKE-1674 and/or viFSP1 (5 μM) and Lip-1 (0.5 μM) for 24 h. j. Cell viability of human cancer cell lines treated with JKE-1674 and/or viFSP1 (5 μM) and Lip-1 (0.5 μM) for 24 h. Data represents the mean ± SD of 3 wells of a 96-well plate from one out of 2 independent experiments (d,f,g,h,i,j).

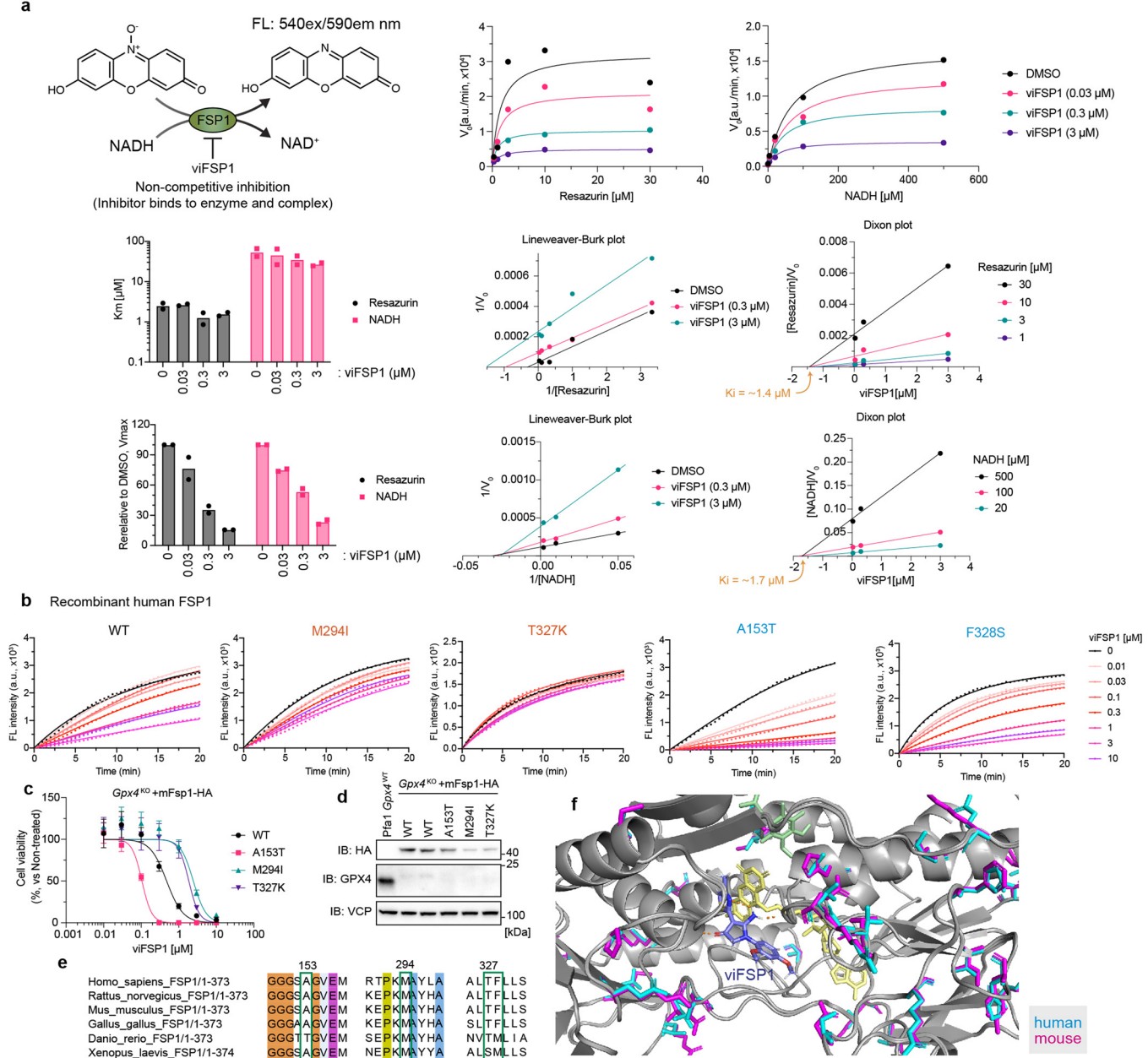

**Extended Data Fig. 7 | Mechanism of action of viFSP1. a.** Enzyme kinetics in vitro assay. Schematic representation of the FSP1 enzyme activity assay using resazurin and NADH as the substrates. Representative reduction kinetics of resazurin, bar plots of the $K_m$ and relative $V_{max}$ values, Lineweaver-Burk plot, and Dixon plot in the presence of different concentrations of either NADH or resazurin with viFSP1 at indicated concentrations. Reduction kinetics, Lineweaver-Burk plot, and Dixon plot represent a single well of a 96-well plate from one out of 2 independent experiments. Bar plots represent the mean from 2 independent experiments. **b.** Representative resazurin reduction assay in the presence of FSP1 mutants with indicated concentrations of viFSP1.

Data represents a single well of a 96-well plate from one out of 3 independent experiments. **c.** Cell viability of Pfa1 *Gpx4*-KO cells stably overexpressing wildtype mouse FSP1 or its mutants treated with viFSP1 for 24 h. Data represents the mean ± SD of 3 wells of a 96-well plate from one out of 2 independent experiments. **d.** Immunoblot analysis of GPX4, FSP1 (HA), and VCP expression in Pfa1 cells stably overexpressing wildtype FSP1 or its mutants from a single experiment. **e.** Alignment of different FSP1 orthologues. **f.** Comparison between predicted human (blue) and mouse (magenta) FSP1 structures in the respective viFSP1 binding pocket. Amino acid differences are highlighted in magenta/blue.

**a** COSMIC analysis

| Position | DNA mutation | AA mutation | Legacy mutation ID | Count |
|---|---|---|---|---|
| 244 | c.731G>A | p.G244D | COSM1492290 | 5 |
| 160 | c.480G>T | p.E160D | COSM4660640 | 2 |
| 160 | c.478G>T | p.E160* | COSM8475894 | 1 |
| 355 | c.1064A>G | p.K355R | COSM5969878 | 1 |
| 285 | c.853G>A | p.D285N | COSM3807587 | 1 |
| 327 | c.980C>T | p.T327M | COSM8368094 | 1 |

**b** DepMap analysis

| Cell Line | Depmap ID | Cancer type | Protein change |
|---|---|---|---|
| SKOV3 | AH-000811 | Ovarian | p.G337D |
| RL952 | AH-000965 | Endometrial/Uterine | p.S6L |

**c** SKOV3 (G337D)

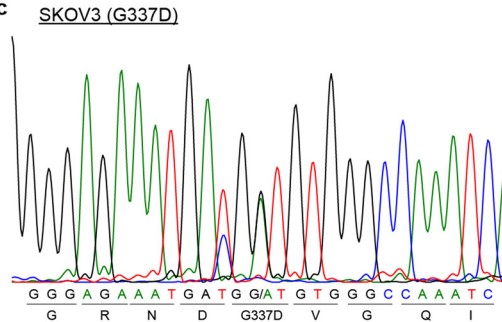

**Extended Data Fig. 8 | Reported *AIFM2* mutations in different cancer types.**
a. COSMIC database analysis. *AIFM2* somatic mutations (heterozygous) are found in cancer patients. Mutations can affect the enzymatic activity of FSP1, such as the proton transfer function (E160, K355), NADH binding (G244), and FAD binding (D285), and may affect the efficacy of FSP1 inhibitors (T327). b. DepMap database analysis. The S6 mutation likely affects FSP1 enzymatic activity via altered myristoylation (consensus motif: MGxxxS). c. Representative sequence data for genomic DNA of *AIFM2* 1 surrounding G337G/D extracted from SKOV3.

# Reporting Summary

## Statistics

For all statistical analyses, confirm that the following items are present in the figure legend, table legend, main text, or Methods section.

| n/a | Confirmed | |
|---|---|---|
| ☐ | ☒ | The exact sample size ($n$) for each experimental group/condition, given as a discrete number and unit of measurement |
| ☐ | ☒ | A statement on whether measurements were taken from distinct samples or whether the same sample was measured repeatedly |
| ☐ | ☒ | The statistical test(s) used AND whether they are one- or two-sided *Only common tests should be described solely by name; describe more complex techniques in the Methods section.* |
| ☒ | ☐ | A description of all covariates tested |
| ☐ | ☒ | A description of any assumptions or corrections, such as tests of normality and adjustment for multiple comparisons |
| ☐ | ☒ | A full description of the statistical parameters including central tendency (e.g. means) or other basic estimates (e.g. regression coefficient) AND variation (e.g. standard deviation) or associated estimates of uncertainty (e.g. confidence intervals) |
| ☐ | ☒ | For null hypothesis testing, the test statistic (e.g. $F$, $t$, $r$) with confidence intervals, effect sizes, degrees of freedom and $P$ value noted *Give P values as exact values whenever suitable.* |
| ☒ | ☐ | For Bayesian analysis, information on the choice of priors and Markov chain Monte Carlo settings |
| ☒ | ☐ | For hierarchical and complex designs, identification of the appropriate level for tests and full reporting of outcomes |
| ☒ | ☐ | Estimates of effect sizes (e.g. Cohen's $d$, Pearson's $r$), indicating how they were calculated |

*Our web collection on statistics for biologists contains articles on many of the points above.*

## Software and code

Policy information about availability of computer code

| Data collection | CytExpert v2.4 (Beckman Coulter), Eve v1.8.2 (Nanolive), Image Lab v6.0 (Biorad), SoftMax Pro v7 (Molecular Devices). |
|---|---|
| Data analysis | Flow Jo v10 software (Treestar, Inc), GraphPad Prism v9 (GraphPad Software), Image J/Fiji v1.53 (NIH), Burrows-Wheeler Alignment Tool (v0.7.17-r1188), MuSyC (https://musyc.lolab.xyz), FAstQC (v0.11.7), Trimmomatic (v039), SAMtools (v1.2), Integrative Genomics Viewer (v2.11.9), AlphaFold2 database (https://alphafold.ebi.ac.uk), seeSAR v12.1 (BioSoveIT), Pymol v2.5.2 (Schrödinger), JalView (v2.11.2.6). |

For manuscripts utilizing custom algorithms or software that are central to the research but not yet described in published literature, software must be made available to editors and reviewers. We strongly encourage code deposition in a community repository (e.g. GitHub). See the Nature Portfolio guidelines for submitting code & software for further information.

## Data

Policy information about availability of data

All manuscripts must include a data availability statement. This statement should provide the following information, where applicable:
- Accession codes, unique identifiers, or web links for publicly available datasets
- A description of any restrictions on data availability
- For clinical datasets or third party data, please ensure that the statement adheres to our policy

All data is available in the Article, the Supplementary Information, and also from the corresponding author on reasonable request. Gel source images are shown in Supplementary Fig. 2. All source data is provided in this paper. Human cancer cell line data was mined from DepMap (https://depmap.org/portal/) or COSMIC

## Human research participants

Policy information about **studies involving human research participants and Sex and Gender in Research.**

| | |
|---|---|
| Reporting on sex and gender | n.a. |
| Population characteristics | n.a. |
| Recruitment | n.a. |
| Ethics oversight | n.a. |

Note that full information on the approval of the study protocol must also be provided in the manuscript.

# Field-specific reporting

Please select the one below that is the best fit for your research. If you are not sure, read the appropriate sections before making your selection.

☒ Life sciences          ☐ Behavioural & social sciences          ☐ Ecological, evolutionary & environmental sciences

For a reference copy of the document with all sections, see nature.com/documents/nr-reporting-summary-flat.pdf

# Life sciences study design

All studies must disclose on these points even when the disclosure is negative.

| | |
|---|---|
| Sample size | Sample sizes in vitro experiments were determined based on the numbers required to achieve statistical significance using indicated statistics, as well as considering of previous publications on similar experiments (PMID: 31634899 and 35922516). |
| Data exclusions | No data exclusions. |
| Replication | The experimental findings were reproduced as validated by at least two independent experiment in Fig. 1-5 and Extended Data Fig. 2-7 except for the mutational analysis. |
| Randomization | Randomization is not relevant to the in vitro experiments since cells come in millions of populations and are automatically randomized and seeded to different wells for treatment. |
| Blinding | In the in vitro experiments, the investigators were not blinded, which is standard in this type of study due to the multiple steps involved that require precise operations for accuracy and precision precluding blinding to experimental variables. |

# Reporting for specific materials, systems and methods

We require information from authors about some types of materials, experimental systems and methods used in many studies. Here, indicate whether each material, system or method listed is relevant to your study. If you are not sure if a list item applies to your research, read the appropriate section before selecting a response.

## Materials & experimental systems

| n/a | Involved in the study |
|---|---|
| ☐ | ☒ Antibodies |
| ☐ | ☒ Eukaryotic cell lines |
| ☒ | ☐ Palaeontology and archaeology |
| ☒ | ☐ Animals and other organisms |
| ☒ | ☐ Clinical data |
| ☒ | ☐ Dual use research of concern |

## Methods

| n/a | Involved in the study |
|---|---|
| ☒ | ☐ ChIP-seq |
| ☐ | ☒ Flow cytometry |
| ☒ | ☐ MRI-based neuroimaging |

## Antibodies

| | |
|---|---|
| Antibodies used | GPX4 (1:1000 for WB, ab125066, Abcam), human FSP1 (1:1000, sc-377120, Santa Cruz Biotechnology), human FSP1(1:5 for WB, clone AIFM2 6D8, rat IgG2a, developed in-house: available from Sigma, |

Cat#MABC1638-25UL), mouse FSP1 (1:100 for WB, clone AIFM2 1A1 rat IgG2a, developed in-house), mouse FSP1(1:5 for WB, clone AIFM2 14D7 IgG2b, developed in-house),  β-actin-HRP (1:50000, A3854, Sigma-Aldrich), valosin containing protein (VCP, 1:10000, ab11433 or ab109240, Abcam), HA tag (YPYDVPDYA, 1:1000 for WB, clone 3F10 rat IgG1, developed in-house) were used in this study. The appropriate secondary antibodies (1:1000-5000, Cell Signaling, Cat#7074S for rabbit; 7076S for mouse, and 1:1000 for anti-rat IgG1b and 2a/b, developed in-house) diluted in 5% skim milk in TBS-T.

**Validation**

GPX4 (ab125066), VCP (ab11433 or ab109240), human FSP1 (sc-377120), HA tag (clone 3F10), β-actin-HRP (A3854) antibodies were validated for WB using mouse and human cell samples in previous publications (PMID: 35922516, 31634899, and 27842070).
FSP1 antibody (clone AIFM2 1A1/6D8 rat IgG2a, and clone AIFM2 14D7 IgG2b, developed in-house) has been validated for WB in previous study (PMID: 35922516).

# Eukaryotic cell lines

Policy information about cell lines and Sex and Gender in Research

**Cell line source(s)**

4-OH-TAM-inducible Gpx4-/- murine immortalized fibroblasts (Pfa1) were established in our lab as reported previously (PMID: 18762024). HT-1080 (CCL-121), HEK293T (CRL-3216), 786-O (CRL-1932), A375 (CRL-1619), B16F10 (CRL-6475), LLC (CRL-1642), MDA-MB-436 (HTB-130), SW620 (CCL-227),  NCI-H460 (HTB-177) and 4T1 (CRL-2539) cells were obtained from ATCC. MC38 cells (SCC172) and SKOV3 (91091004) were obtained from Sigma-Aldrich. HEC151 cells (JCRB1122-A) were obtained from Tebubio. Rat1 cells (available from Thermo Fisher) were kindly gifted from Medizinische Hochschule Hannover. Huh7 cells (available from Thermo Fisher) were kindly gifted from Dr. Robert Schneider, Helmholtz Munich. MC38 cells (available from Sigma) were kindly gifted from Dr. Patrizia Agostinis (KU Leuven, Belgium).

**Authentication**

None of the cell lines used were authenticated.

**Mycoplasma contamination**

All cell lines were tested negative for mycoplasma contamination.

**Commonly misidentified lines**
(See ICLAC register)

No commonly misidentified cell lines were used.

# Flow Cytometry

## Plots

Confirm that:

☒ The axis labels state the marker and fluorochrome used (e.g. CD4-FITC).

☒ The axis scales are clearly visible. Include numbers along axes only for bottom left plot of group (a 'group' is an analysis of identical markers).

☒ All plots are contour plots with outliers or pseudocolor plots.

☒ A numerical value for number of cells or percentage (with statistics) is provided.

## Methodology

**Sample preparation**

100,000 cells per well were seeded on a 12-well plate one day prior to the experiments. On the next day, cells were treated with 2.5 μM icFSP1 for 3 h, and then incubated with 1.5 μM C11-BODIPY 581/591 (Invitrogen, Cat#D3861) for 30 min in a 5% CO2 atmosphere at 37°C. Subsequently, cells were washed by PBS once and trypsinized, and then resuspended in 500 μL PBS. Cells were passed through a 40 μm cell strainer and analyzed by a flow cytometer (CytoFLEX, Beckman Coulter) with a 488-nm laser for excitation. Data was collected from the FITC detector (for the oxidized form of BODIPY) with a 525/40nm bandpass filter and from the PE detector (for the reduced form of BODIPY) with a 585/42 nm bandpass filter.

**Instrument**

CytoFLEX (Beckman Coulter)

**Software**

CytExpert v2.4 was used for data collection. FlowJo v10 was used for data analysis.

**Cell population abundance**

At least 10,000 cells were analyzed for each sample.

**Gating strategy**

Cell populations were separated from cellular debris using FSC and SSC.

☒ Tick this box to confirm that a figure exemplifying the gating strategy is provided in the Supplementary Information.

