## [Peer Review File · Nature Structural & Molecular Biology]

Peer Review Information

Manuscript Title: Integrated chemical and genetic screens unveil FSP1 mechanisms of ferroptosis regulation

Corresponding author name(s): Marcus Conrad

Reviewer Comments & Decisions:

Decision Letter, initial version:

Message: 25th Apr 2023

Dear Dr. Conrad,

Thank you again for submitting your manuscript "Integrated chemical-genetic screens unveil FSP1 mechanisms and ferroptosis vulnerabilities". We now have comments (below) from the 3 reviewers who evaluated your paper. In light of those reports, we remain interested in your study and would like to see your response to the comments of the referees, in the form of a revised manuscript.

You will see that overall, all referees are positive about the study and its novel conclusions. While Reviewers #1 and #3 have minor concerns that should be addressed with clarifications and textual comments, Reviewer #2 had raised some important points, that should be addressed with further experimental data. After discussion with the editorial team, we agree that expanding the analysis to additional cancer cell lines that have high expression of FSP1, would be important to increase the impact of the study, in addition to addressing the other technical comments.

We are committed to providing a fair and constructive peer-review process. Do not hesitate to contact us if there are specific requests from the reviewers that you believe are technically impossible or unlikely to yield a meaningful outcome. We expect to see your revised manuscript within 6 weeks. If you cannot send it within this time, please contact us to discuss an extension; we would still consider your revision, provided that no similar work has been accepted for publication at NSMB or published elsewhere.

As you already know, we put great emphasis on ensuring that the methods and statistics reported in our papers are correct and accurate. As such, if there are any changes that should be reported, please submit an updated version of the Reporting Summary along

with your revision.

Reporting Summary:

Please note that all key data shown in the main figures as cropped gels or blots should be presented in uncropped form, with molecular weight markers. These data can be aggregated into a single supplementary figure item. While these data can be displayed in a relatively informal style, they must refer back to the relevant figures. These data should be submitted with the final revision, as source data, prior to acceptance, but you may want to start putting it together at this point.

SOURCE DATA: we request that authors provide, in tabular form, all the data underlying the graphical representations used in figures. This is to further increase transparency in data reporting, as detailed in this editorial (<http://www.nature.com/nsmb/journal/v22/n10/full/nsmb.3110.html>). Spreadsheets can be submitted in excel format. Only one (1) file per figure is permitted; thus, for multi-paneled figures, the source data for each panel should be clearly labeled in the Excel file; alternately the data can be provided as multiple, clearly labeled sheets in an Excel file. When submitting files, the title field should indicate which figure the source data pertains to. We request our authors to provide source data at the revision stage, so that they are part of the peer-review process. Please also include the uncropped blots in the Source data file.

Data availability: this journal strongly supports public availability of data. All data used in accepted papers should be available via a public data repository, or alternatively, as Supplementary Information. If data can only be shared on request, please explain why in

your Data Availability Statement, and also in the correspondence with your editor. Please note that for some data types, deposition in a public repository is mandatory - more information on our data deposition policies and available repositories can be found below: <https://www.nature.com/nature-research/editorial-policies/reporting-standards#availability-of-data>

[redacted]

Sincerely,

Carolina Perdigoto, PhD
Chief Editor
Nature Structural & Molecular Biology
orcid.org/0000-0002-5783-7106

Referee expertise:

Referee #1: Ferroptosis and inflammation in disease

Referee #2: FSP1/ferroptosis in cancer

Referee #3: Cancer biology and drug discovery

Reviewers' Comments:

Reviewer #1:

Remarks to the Author:

Nakamura et al. discovered mode of action of FSP1 in relation to its redox activity/functioning and pharmacological targeting. In addition, they identified species independent inhibitor of FSP1.

This is overall well-designed study using innovative approaches to elucidate MoA of FSP1. The identification of crucial AA's for its function or targeting will be of utmost importance in context of ferroptosis directed anti-cancer therapies. The conclusions are supported by the data presented. I only have some minor issues:

- viFSP1 treatment as such seems not toxic in most cancer cell lines in contrast to e.g. GPX4 targeting strategies. This implies that FSP1 targeting is inferior to GPX4 in relation to unleashing ferroptosis brakes, which is of importance for designing future ferroptosis anti-cancer approaches. Please elaborate on this in the text. A synergistic effect is mentioned; however, this is hard to judge on heatmaps. Please clarify.
- In the discussion, authors hint in direction of precision oncology viz. somatic mutation that might predict sensitivity or resistance to FSP1-directed ferroptosis anti-cancer therapies. However, considering important but inferior role of FSP1 compared to GPX4, how do the authors see this practically in relation to ferroptosis induction in specific cancers? Should FSP1 targeting be considered only when GPX4 expression is low? Or cotargeting of FSP1 inhibitors with GPX4-targeting strategies in tumors with high GPX4? Please clarify.
- Structural models are not experimental, and should be interpreted with caution.

Reviewer #2:

Remarks to the Author:

In this manuscript by Nakamura et al., the authors identify the binding sites and mechanism of a human-selective inhibitor of ferroptosis suppressor protein 1 (FSP1) and in doing so, reveal novel active and inactive FSP1 mutants and present a novel FSP1 inhibitor (viFSP1) applicable in various species. Given the tremendous interest in targeting the ferroptosis pathway in treatment resistant cancer, this is a very relevant and important study to add to this arsenal. The authors perform several error-prone PCR mutagenesis screens to identify gain and loss of function FSP1 mutants and track down the inhibitor binding sites. Overall, the study presents very solid biochemical work, only a few additional points require consideration:

Given the important implications for their finding's applicability in cancer, screening for

potential synergy in cancer cell lines should be extended and in particular include cancer cell lines with validated high expression/upregulation of FSP1. As is, the cell lines used in Extended Data Figure 5d-f in many cases do not show synergy in 3 out of 6 cases and the reader is left wondering whether they just don't express FSP1 in the first place.

In figure 3, the authors should provide experiments in which they attempt to humanize mFSP1 in F360 and possibly other sites to test whether this is sufficient to sensitize to iFSP1 upon mFSP1 overexpression.

In figure 5, the authors identify viFSP1 resistant and sensitive mutations in hFSP1. How do these sites look in mFSP1? Given that mFSP1 is sensitive to viFSP1 one would expect that those sites are conserved between human and mouse. Can the authors elaborate on this and test one or the other mutant also in mFSP1 treated with viFSP1?

Throughout the manuscript, the authors should consider including other more selective GPX4 inhibitors than RSL3 in several of the key experiments.

Given their scope, prior work on FSP1 in cancer should be included in the citations

Reviewer #3:

Remarks to the Author:

The authors build on their previous discovery that FSP1 is a key repressor of ferroptosis. Thus, inhibiting FSP1 is expected to trigger ferroptosis. Prior work identified an inhibitor of FSP1 that can induce ferroptosis in cancer cells. Developing ferroptosis inducers to eradicate specific populations of cancer cells is meaningful. It is not clear how this inhibitor binds to FSP1 and how it exerts its activity. The authors have now established a robust methodology combining genetic and chemical screens to identify key residues of FSP1 that are important for its functions and which can affect drug target engagement. Furthermore, the authors have identified new structures that can target FSP1 across species, providing mechanistic insights that could not be obtained with previous inhibitors. Understanding how small molecule interact with their target is important as it might guide the design of other drug-like molecules and can inform on how these drugs could be used in clinical settings (e.g. cancer mutations). The methodology is sound overall. The result support the conclusions. The argument of targeting FSP1 over GPX4 stating that FSP1 is not essential is compelling. Thank you for pointing this out. The manuscript is clear for the most part, difficult to read at times. The authors may want to consider the following points in a revised version of their manuscript.

1. I would suggest another title like: 'Integrated chemical and genetic screens reveal FSP1 mechanisms of ferroptosis regulation'

chemical-genetic screens gives the impression of a dual/combined screen whereas if I understood properly, the authors have performed both types of screens independently in their study. Also 'mechanisms' alone is vague. Mechanisms of what? Last, this paper is about FSP1, and does not provide new insights on ferroptosis vulnerabilities per se. The title as is can be misleading.

2. In the abstract, I would remove 'difficult-to-treat'. All cancers are difficult to treat. Some are incurable. Perhaps 'eradicate specific cancer cell populations refractory to

standard-of-care...

3. 'by studying somatic mutations occurring in cancer, by performing'. This sentence is too long and unclear.

4. 'proton transfer pathway'. This is a function or a mechanism. Not a pathway.

5. The authors claim species-independent FSP1 inhibitors but only two species have been experimentally explored. Is this an appropriate wording?

6. 'intriguing insights'. 'New insights' perhaps? What is intriguing?

7. 'rationale design'. 'Rational' !

8. No need to define ferroptosis as a non-apoptotic cell death. In that case, it is a non-many things. Shall there all be listed. Just define ferroptosis for what it is. Not what it is not.

9. In the abstract: 'targeting certain cancer 'CELL' state.

10. (i.e., iFSP16). The ref may be after the).

11. a single mutated plasmid: rephrase with 'plasmid encoding single mutated FSP1'. Not sure the plasmids are mutated. They contain the inserted mutations do they not?

12. CoQ1013. The ref may be listed at the end of the sentence for clarity.

13. Lines 107 and 199. 'confirming'. Without X-ray or cryoEM data, it should perhaps read 'supporting the contention that'...

14. Line 250. 'Concluded' same comment.

15. Line 157/158 is difficult to read.

16. Line 108. 'Allowing each cell to express a single mutant'. Is this correct, or an assumption based on the experimental conditions?

17. Line 231: non-competitive inhibitors. I may have missed something as the authors define binding in the quinone binding site and NADH binding site. Please clarify.

18. Can amino acid replacement impact on FSP1 folding/subcellular localization and thus, drug target engagement (instead of directly altering the binding site)? If so, this may be discussed explicitly.

19. RSL3 chemically reacts with cysteine, just like the selenide of GPX4. While it might be a good ferroptosis inducer, I am not sure one should claim it is a GPX4 inhibitor. One may expect reactions with glutathione. To be confirmed.

Two other comments:

21. Targeting cancer cell states is not shown in this paper. Some phrasing in the paper

may wrongly convey that information (see current title).

22. Refs 2-5 are fine. The authors may consider Mai et al. Nature Chem. 9, 1025-1033 (2017) and Muller et al. Nature Chem. 12, 929-938 (2020), which actually preceded refs 2-5 and provide evidence that the iron load is higher in the mesenchymal (pro-metastatic) cancer cell state. Since iron is the ferroptosis trigger, it would make sense to take these references into account. Besides, refs 2-5 do not discuss the higher iron load in cancer cell states.

Author Rebuttal to Initial comments

NSMB-A47433-T – Revision 1

We thank all reviewers and the editor for the critical assessment of our manuscript and the highly appreciative comments made by the reviewers. Please find below our response to each comment on a point-by-point basis. Please also find the changes in the text highlighted in this letter as underlined and in the revised manuscript as marked in red.

Reviewers' Comments:

Reviewer #1:

Remarks to the Author:

Nakamura et al. discovered mode of action of FSP1 in relation to its redox activity/functioning and pharmacological targeting. In addition, they identified species independent inhibitor of FSP1.

This is overall well-designed study using innovative approaches to elucidate MoA of FSP1. The identification of crucial AA's for its function or targeting will be of utmost importance in context of ferroptosis directed anti-cancer therapies. The conclusions are supported by the data presented. I only have some minor issues:

We are thankful for the highly appreciative comments and are very much pleased that the reviewer shares her/his enthusiasm for our work.

- viFSP1 treatment as such seems not toxic in most cancer cell lines in contrast to e.g GPX4 targeting strategies. This implies that FSP1 targeting is inferior to GPX4 in relation to unleashing ferroptosis brakes, which is of importance for designing future ferroptosis anti-cancer approaches. Please elaborate on this in the text. A synergistic effect is mentioned; however, this is hard to judge on heatmaps. Please clarify.

We agree that targeting FSP1 alone is generally not sufficient to induce ferroptosis as compared to GPX4. However, GPX4 is known to be essential for early embryogenesis and tissue homeostasis of several organs. This suggests that strategies to robustly inhibit or degrade GPX4 alone are likely associated with detrimental off-target effects, thereby severely limiting the therapeutic window of these approaches. In contrast, FSP1 knockout (KO) mice (*Aifm2^{tm1Marc}*) are fully viable and display no phenotype at least under normal housing conditions (see Tonnus *et al.* Nat Commun 2021, PMID: 34285231; Mishima *et al.* Nature 2022, PMID: 35922516), whereas tamoxifen-inducible whole body GPX4 KO mice (except in brain) display acute kidney failure (see Angeli *et al.* Nat. Cell Biol.,

2014, PMID: 25402683), implying that targeting GPX4 can show severe off-target effects other than the tumor. Besides, several types of neurons, certain immune cells, hepatocytes and endothelial cells show a strong dependence on GPX4 (Conrad *et al.* Trends Mol Med 2021, PMID: 32958404). Thus, GPX4 inhibitors should be ideally delivered in a cancer cell type- and site-specific manner at carefully controlled concentrations, whereas FSP1 inhibitors are perhaps more straightforward to be used. Therefore, we still deem FSP1 as a very attractive target for cancer treatment with limited off-target effects if we can target FSP1 in synergy with GPX4 specifically in tumors.

Regarding this discussion, we added the text in the main text on page 10 as follows;

Given that some cancer cells are resistant to GPX4 inhibition-induced ferroptosis and FSP1 inhibitors sensitize a number of cancer cells to ferroptosis induced by sublethal GPX4 inhibition, the combination therapy of FSP1 inhibitors with canonical ferroptosis inducers, such as GPX4 and system X_c inhibitors, ideally a tumor-specific manner, could be a valid new potential anticancer therapy.

Per the reviewer's suggestion, we additionally assessed the synergistic effect of viFSP1 with the GPX4 inhibitor (RSL3) using multidimensional synergy of combinations (MuSyC) calculation as illustrated in Extended Data Fig. 5e-g. Based on this calculation, viFSP1 shows highly synergistic potency and efficacy, in particular in RSL3-resistant cancer cell lines, such as B16F10, A375 and H460 cells, which display high expression of FSP1 (Extended Data Fig. 6a-c).

Extended Data Fig. 5

Extended Data Fig. 6

• In the discussion, authors hint in direction of precision oncology viz. somatic mutation that might predict sensitivity or resistance to FSP1-directed ferroptosis anti-cancer therapies. However, considering important but inferior role of FSP1 compared to GPX4, how do the authors see this practically in relation to ferroptosis induction in specific cancers? Should FSP1 targeting be considered only when GPX4 expression is low? Or cotargeting of FSP1 inhibitors with GPX4-targeting strategies in tumors with high GPX4? Please clarify.

We thank the reviewer for bringing this up. Given that some cancer cell lines are highly resistant to GPX4 inhibition alone, such as H460 cells with low GPX4 expression or MDA-MB-436 cells with

both high FSP1 and GPX4 expression (Extended Data Fig. 6c), co-treatment of ferroptosis inducers with FSP1 inhibitors would be a promising approach when aiming to target these ferroptosis-resistant cancer cells. Moreover, as outlined in the foregoing efficient killing of tumors by GPX4 alone might be associated with severe side effects due to GPX4's essential role for many tissues and organs.

To support the co-targeting of FSP1 and GPX4, we tested numerous cancer cell lines with FSP1 expression, and most of them showed a synergistic effect of FSP1 inhibition with GPX4 inhibitors as now shown in new Extended Data Fig. 6d.

Extended Data Fig. 6

Based on this discussion we highlighted the importance of co-targeting FSP1 with GPX4 in the discussion on page 10:

Given that some cancer cells are resistant to GPX4 inhibition-induced ferroptosis and FSP1 inhibitors sensitize a number of cancer cells to ferroptosis induced by sublethal GPX4 inhibition, the combination therapy of FSP1 inhibitors with canonical ferroptosis inducers, such as GPX4 and system Xc inhibitors ideally a tumor-specific manner, could be a valid new potential anticancer therapy.

- Structural models are not experimental, and should be interpreted with caution.

We fully agree with your comment. While we are intensively trying to elucidate the 3-D structure of FSP1 using all established state-of-the-art methods in the field, our attempts have failed so far to generate the structure of FSP1 but might be successful in the near future using a different approach.

As also suggested by reviewer#3, we toned down the sentences based on the predicted structure using AlphaFold2. We now add a sentence to point out the limitation of our study in the interpretation of the superimposed FSP1 structure (see discussion on page 12).

As shown above, this kind of information can be very useful; however, caution needs to be taken into account when interpreting data based on modeling approaches, calling for an experimentally validated 3-D structure of FSP1 in the near future.

Reviewer #2:

Remarks to the Author:

In this manuscript by Nakamura et al., the authors identify the binding sites and mechanism of a human-selective inhibitor of ferroptosis suppressor protein 1 (FSP1) and in doing so, reveal novel active and inactive FSP1 mutants and present a novel FSP1 inhibitor (viFSP1) applicable in various species. Given the tremendous interest in targeting the ferroptosis pathway in treatment resistant cancer, this is a very relevant and important study to add to this arsenal. The authors perform several error-prone PCR mutagenesis screens to identify gain and loss of function FSP1 mutants and track down the inhibitor binding sites. Overall, the study presents very solid biochemical work, only a few additional points require consideration:

We are very pleased that we can draw the interest of the reviewer and that she/he shares great enthusiasm for our work.

Given the important implications for their finding's applicability in cancer, screening for potential synergy in cancer cell lines should be extended and in particular include cancer cell lines with validated high expression/upregulation of FSP1. As is, the cell lines used in Extended Data Figure 5d-f in many cases do not show synergy in 3 out of 6 cases and the reader is left wondering whether they just don't express FSP1 in the first place.

Thank you for the suggestion. We assessed the synergistic effect of viFSP1 with the GPX4 inhibitor (RSL3) using the multidimensional synergy of combinations (MuSyC) calculation in Extended Data Fig 5d-g. Based on this calculation, viFSP1 shows highly synergistic potency and efficacy, in particular in RSL3-resistant cancer cell lines, such as B16F10, A375 and H460 cells, with high expression of FSP1 (Extended Data Fig. 6). Regarding other RSL3-sensitive cell lines, such as Rat1 and 786-O cells, it will be difficult to see the synergistic effect of viFSP1 in our tested concentrations of RSL3 when the cells *per se* are already highly sensitive to RSL3, even though FSP1 expression is confirmed (Extended Data Fig. 6c). Besides, we screened FSP1 expression in cancer cell lines using the database (<https://depmap.org/portal/> or <http://tismo.cistrome.org>), and most cancer cells express FSP1, albeit to varying degrees. Thus, we assume the synergistic effect will always depend on the intrinsic sensitivity of cells toward ferroptosis and the inducers' concentrations in each cancer cell. Moreover, there are other enzymatic and non-enzymatic ferroptosis surveillance systems such as GCH1, BH2/BH4, squalene, and 7-DHC etc. that will have an impact on the

sensitivity of cells toward ferroptosis, which may also depend on the origin of the cancer cell *per se*.

Extended Data Fig. 5

Extended Data Fig. 6

Besides, we tested additional cell lines with validated FSP1 expression to see if viFSP1 can sensitize cells to GPX4 inhibitors in Extended Data Fig. 6d.

Extended Data Fig. 6

In figure 3, the authors should provide experiments in which they attempt to humanize mFSP1 in F360 and possibly other sites to test whether this is sufficient to sensitize to iFSP1 upon mFSP1 overexpression.

We are thankful for bringing this up. As suggested by the referee, we generated mFSP1 L360F mutants and tested the sensitivity toward iFSP1. This “humanized mFSP1 L360F” indeed became sensitive to iFSP1, while mFSP1 is resistant as shown in new Fig. 3c. This is similar to what was recently reported by Nishida Xavier da Silva *et al.* Cell Death Dis 2023.

Fig. 3
C Pfa1 *Gpx4*^{KO} + FSP1-HA

In figure 5, the authors identify viFSP1 resistant and sensitive mutations in hFSP1. How do these sites look in mFSP1? Given that mFSP1 is sensitive to viFSP1 one would expect that those sites are

conserved between human and mouse. Can the authors elaborate on this and test one or the other mutant also in mFSP1 treated with viFSP1?

As suggested by the referee, we included the differences in sequence and structure between human and mouse FSP1 in new Extended Data Fig. 7c-f. Besides, we generated mFSP1 A153T, M294I, and T327K mutants and tested their sensitivity toward viFSP1. The sensitivity of these mutants toward viFSP1 was similar to the human mutants.

Extended Data Fig. 7

Throughout the manuscript, the authors should consider including other more selective GPX4 inhibitors than RSL3 in several of the key experiments.

We thank the referee for this suggestion. We tested JKE-1674, which is considered to be the next-generation GPX4 inhibitor and active metabolite of ML210 for some cell lines (see Eaton *et al.* Nat Chem Biol 2020; PMID: 32231343). Moreover, given that almost all GPX4 inhibitors ultimately target the active site of GPX4 (i.e., selenolate) and that we cannot formally exclude the possibility that any available GPX4 inhibitors would target also other selenoproteins (as reported by Chen *et al.* J Am Chem Soc 2018; PMID: 29569437), we tested the sensitivity of viFSP1 toward other murine and human cancer cell lines, in which GPX4 was genetically deleted and therefore they rely on the expression of human or mouse FSP1. All these data indicate that viFSP1 enhances ferroptosis sensitivity induced by GPX4 inhibition or deletion (see new Extended Data Fig. 6e-j).

Extended Data Fig. 6

Given their scope, prior work on FSP1 in cancer should be included in the citations

Thank you for the suggestion. We include more citations in the revised version that link FSP1 and cancer (Pontel *et al.* Redox Biol., 2022, PMID: 35944469; Müller *et al.* Cell Death Differ 2023, PMID: 35459868; Koppula *et al.* Nat Commun 2022, PMID: 35459868).

Therefore, targeting FSP1 in tumor cells¹⁴⁻¹⁶ might be preferred in contrast to GPX4, which is known to be essential for early embryogenesis and tissue homeostasis in a variety of organs such as kidney, liver and brain¹⁷.

Reviewer #3:

Remarks to the Author:

The authors build on their previous discovery that FSP1 is a key repressor of ferroptosis. Thus, inhibiting FSP1 is expected to trigger ferroptosis. Prior work identified an inhibitor of FSP1 that can induce ferroptosis in cancer cells. Developing ferroptosis inducers to eradicate specific populations of cancer cells is meaningful. It is not clear how this inhibitor binds to FSP1 and how it exerts its activity. The authors have now established a robust methodology combining genetic and chemical screens to identify key residues of FSP1 that are important for its functions and which can affect drug target engagement. Furthermore, the authors have identified new structures that can target FSP1 across species, providing mechanistic insights that could not be obtained with previous inhibitors. Understanding how small molecule interact with their target is important as it might guide the design of other drug-like molecules and can inform on how these drugs could be used in clinical settings (e.g. cancer mutations). The methodology is sound overall. The result support the conclusions. The argument of targeting FSP1 over GPX4 stating that FSP1 is not essential is compelling. Thank you for pointing this out. The manuscript is clear for the most part, difficult to read at times. The authors may want to consider the following points in a revised version of their manuscript.

We are very pleased to read that the reviewer shares the passion for the topic and values our work. We greatly appreciate the many constructive comments made on our manuscript.

1. I would suggest another title like: 'Integrated chemical and genetic screens reveal FSP1 mechanisms of ferroptosis regulation'

chemical-genetic screens gives the impression of a dual/combined screen whereas if I understood properly, the authors have performed both types of screens independently in their study. Also 'mechanisms' alone is vague. Mechanisms of what? Last, this paper is about FSP1, and does not provide new insights on ferroptosis vulnerabilities per se. The title as is can be misleading.

We apologize for the title which may create some sort of confusion or be misleading. Per the reviewer's suggestion, we changed the title as follows:

"Integrated chemical and genetic screens unveil FSP1 mechanisms of ferroptosis regulation."

2. In the abstract, I would remove 'difficult-to-treat'. All cancers are difficult to treat. Some are incurable. Perhaps 'eradicate specific cancer cell populations refractory to standard-of-care...

We agree and removed 'difficult-to-treat' from the abstract.

3. 'by studying somatic mutations occurring in cancer, by performing'. This sentence is too long and unclear.

We accordingly changed the sentence in the abstract as follows:

Since its molecular mechanisms have remained obscure, we studied numerous FSP1 mutations present in cancer or identified by untargeted random mutagenesis.

4. 'proton transfer pathway'. This is a function or a mechanism. Not a pathway.

Per the referee's suggestion, we changed the sentence in the abstract as follows;

This mutational analysis elucidates the FAD/NAD(P)H binding site and proton transfer mechanism of FSP1, which emerged to be evolutionarily conserved among different NADH quinone reductases.

5. The authors claim species-independent FSP1 inhibitors but only two species have been experimentally explored. Is this an appropriate wording?

In our manuscript, we overexpressed 4 different species of FSP1 in Pfa1 cells (i.e., human, mouse, rat, and chicken, please see Extended Data Fig. 5b). Besides, all mutations resistant to viFSP1 are conserved to some extent, and thus we believe that viFSP1 can be a species-independent FSP1 inhibitor.

6. 'intriguing insights'. 'New insights' perhaps? What is intriguing?

We accordingly changed the sentence in the abstract as follows;

Conclusively, our study provides new insights into the molecular functions of FSP1 and enables the rational design of FSP1 inhibitors targeting certain cancer cells.

7. 'rationale design'. 'Rational' !

As per this suggestion, we changed the sentence in the abstract as follows;

Conclusively, our study provides new insights into the molecular functions of FSP1 and enables the rational design of FSP1 inhibitors targeting certain cancer cells.

8. No need to define ferroptosis as a non-apoptotic cell death. In that case, it is a non-many things. Shall there all be listed. Just define ferroptosis for what it is. Not what it is not.

Thanks for bringing this up. The main text now reads as follows:

Since the recognition as a distinct iron-dependent cell death characterized by the oxidative destruction of cellular membranes, ferroptosis has attracted tremendous interest likely owing to its high relevance in human diseases such as neurodegenerative disorders, tissue ischemia-reperfusion injury and malignancies¹

9. In the abstract: 'targeting certain cancer 'CELL' state.

As per this suggestion, we changed the sentence in the abstract as follows;

Conclusively, our study provides new insights into the molecular functions of FSP1 and enables the rational design of FSP1 inhibitors targeting certain cancer cells.

10. (i.e., iFSP16). The ref may be after the).

We changed the sentence in the text as follows:

Besides, since the first reported inhibitor for FSP1 (i.e., iFSP1)⁸ is specific for the human enzyme^{18, 19}, its species-specificity precludes in-depth studies on its precise mechanism-of-action (MoA), thus hindering the analysis of organismal differences and similarities among different FSP1 orthologues.

11. a single mutated plasmid: rephrase with 'plasmid encoding single mutated FSP1'. Not sure the plasmids are mutated. They contain the inserted mutations do they not?

Thanks for the suggestion and for having a keen eye. We changed the sentence as follows;

This plasmid pool was then transduced into Pfa1 cells with a very low infection ratio (MOI = approx. 0.1), in analogy to ours' and others' genome-wide CRISPR screening approaches²⁹⁻³¹, allowing each cell to express a plasmid with a single mutated FSP1.

12. CoQ1013. The ref may be listed at the end of the sentence for clarity.

Per suggestion, we changed the sentence as follows;

To further validate these consensus motifs, we took advantage of the predicted protein structure of FSP1 derived from AlphaFold2^{26,27}, which perfectly aligns with the crystal structure of the yeast NDH-2 enzyme known as Ndi1²⁸ (Extended Data Fig. 1b) resulting in the superimposed, modeled structure of FSP1 with putative binding sites of FAD, NADH and CoQ₁₀ (Extended Data Fig. 2g)¹⁸.

13. Lines 107 and 199. 'confirming'. Without X-ray or cryoEM data, it should perhaps read 'supporting the contention that'...

We accordingly changed the sentence as follows:

According to this modeled FSP1 structure, G244 is an essential constituent of the NADH binding domain (Fig. 1g) similar to other NDH-2 (Extended Data Fig. 1b), further supporting the findings that the somatic mutation G244D abrogates the ferroptosis-suppressive role of FSP1.

In light of the fact that these mutants are faced to the expected membrane-attaching surface and quinone binding pocket (Extended Data Fig. 2g), it is assumed that iFSP1 targets the quinone binding site (Fig. 3g)^{18,19}.

14. Line 250. 'Concluded' same comment.

Per the reviewer's suggestion, we changed the sentence as follows;

Since these amino acids are located in the NAD(P)H binding site and are highly conserved among species (Extended Data Fig. 7c-f), it can be assumed that viFSP1 targets the NADH binding pocket (Fig. 5d).

15. Line 157/158 is difficult to read.

Thanks for the suggestion; we changed the sentence as follows;

Thus, this highly conserved proton transfer mechanism involving sequential carboxylic acid and lysine residues is most likely crucial for the quinone protonation function of FSP1.

16. Line 108. 'Allowing each cell to express a single mutant'. Is this correct, or an assumption based on the experimental conditions?

This is anticipated from the established methodology in CRISPR screen (i.e., low MOI), where we assume that cells express a plasmid with a single mutated FSP1.

17. Line 231: non-competitive inhibitors. I may have missed something as the authors define binding in the quinone binding site and NADH binding site. Please clarify.

There are 3 modes of competitive inhibition in light of the established simplified biochemistry models: competitive, non-competitive, and uncompetitive. To define these modes, different concentrations of the substrate should be added to the system and the inhibitory mechanisms of inhibitors can be predicted by the shape of the reaction curve (the Lineweaver-Burk and Dixon plots). From these two plots, we may estimate the mode of FSP1 inhibitors as the non-competitive model, that is, the binding model of inhibitors targeting both enzyme (substrate-free) and complex (enzyme-substrate binding). However, these models might not be suitable for FSP1 because of its complexity in redox reaction (i.e., different 3 substrates and their complex redox state in FSP1).

Considering this assumption and limitation, we tone down the sentence as follows:

From the Lineweaver-Burk and Dixon plots³⁹, it can be speculated that iFSP1 and viFSP1 are both non-competitive inhibitors, which means these FSP1 inhibitors can bind either the enzyme in the presence or absence of substrates; however, this should be experimentally investigated when the 3-D structure of FSP1 becomes available in the future.

18. Can amino acid replacement impact on FSP1 folding/subcellular localization and thus, drug target engagement (instead of directly altering the binding site)? If so, this may be discussed explicitly.

We fully agree with this point. We did not test all mutants for proper structure/folding and localization.

However, all inhibitor-resistant mutants presented in our manuscript afford survival of cells after GPX4 deletion in Pfa1 cells. Thus, we expect that all FSP1 mutants can be functional and active at the membrane and reduce extramitochondrial CoQ₁₀, otherwise the cells would die like with the membrane binding-defective mutant, G2A.

As already observed with the band shift in the western blot in FSP1 E156A and E160A, we cannot exclude the possibility of changes in folding and perhaps other yet-unrecognized post-translational modifications.

Based on the referee's suggestion, we describe this possibility in the discussion on page 11-12 as follows:

We confirmed that all mutants resistant/sensitive to FSP1 inhibitors indeed afford survival of cells after genetic deletion of Gpx4, suggesting that these mutants must be functional. However, as experimental data on the 3-D structures of FSP1 remain elusive at this stage, we cannot formally exclude the possibility that amino acid mutations can impact on folding and/or other yet-unrecognized post-translational modifications of FSP1. As shown above, this kind of information can be very useful; however, caution needs to be taken into account when interpreting data based on modeling approaches, calling for an experimentally validated 3-D structure of FSP1 in the near future.

19. RSL3 chemically reacts with cysteine, just like the selenide of GPX4. While it might be a good ferroptosis inducer, I am not sure one should claim it is a GPX4 inhibitor. One may expect reactions with glutathione. To be confirmed.

Given the scope of our paper investigating the reduction mechanism by FSP1 and the mechanism of action of FSP1 inhibitors, we cannot provide any information about the specificity of GPX4 inhibitors. Although it is known that RSL3 targets the selenolate residue of almost all selenoproteins (Chen *et al.* JACS, 2018 (PMID: 29569437)) at least in the cellular context, it does also target the cysteine in GPX4 as reported by Yang *et al.* PNAS, 2016 (PMID: 27506793). Yet, RSL3-induced ferroptosis is mainly induced by its binding to selenolate of GPX4 (U46) as shown in Yang *et al.* PNAS, 2016 (PMID: 27506793) and Ingold *et al.* Cell, 2018. In the range of "normal" working concentrations (< 3 μ M), RSL3-induced ferroptosis can be rescued in Gpx4 KO cells overexpressing Gpx4 U46C, but not WT (see also below) (Ingold *et al.* Cell, 2018 (PMID: 29290465)). Thus, although RSL3 can chemically react with cysteine in cells, we infer that RSL3 is still a valid GPX4 inhibitor to induce ferroptosis, targeting the selenolate of GPX4 at the given concentration range used in this study (0-3 μ M).

g

Figures from Ingold, et al, Cell, 2018

Two other comments:

21. Targeting cancer cell states is not shown in this paper. Some phrasing in the paper may wrongly convey that information (see current title).

We apologize for this misconception - we accordingly amended the title. Please see above.

22. Refs 2-5 are fine. The authors may consider Mai et al. Nature Chem. 9, 1025-1033 (2017) and Muller et al. Nature Chem. 12, 929-938 (2020), which actually preceded refs 2-5 and provide evidence that the iron load is higher in the mesenchymal (pro-metastatic) cancer cell state. Since iron is the ferroptosis trigger, it would make sense to take these references into account. Besides, refs 2-5 do not discuss the higher iron load in cancer cell states.

Thank you for the great suggestion. We include both papers in the revised version and describe the connection between iron content and ferroptosis in cancer cells in the introduction.

In particular, certain cancer cell states, including cancer stem cells, therapy-resistant and disseminating cancer cells have been reported to exhibit an inherent vulnerability to ferroptosis, providing the rationale for selectively inducing ferroptosis as a next-generation anticancer therapy approach²⁻⁷.

Decision Letter, first revision:

Message: Our ref: NSMB-A47433A

20th Jul 2023

Dear Dr. Conrad,

Thank you for submitting your revised manuscript "Integrated chemical and genetic screens unveil FSP1 mechanisms of ferroptosis regulation" (NSMB-A47433A). It has now been seen by the original referees and their comments are below. The reviewers find that the paper has improved in revision, and therefore we'll be happy in principle to publish it in Nature Structural & Molecular Biology, pending minor revisions to satisfy our editorial and formatting guidelines.

Sincerely,

Carolina Perdigoto, PhD
Chief Editor
Nature Structural & Molecular Biology
orcid.org/0000-0002-5783-7106

Reviewer #2 (Remarks to the Author):

The authors have addressed all points raised at the highest possible standards.

Reviewer #3 (Remarks to the Author):

Nice revision. The authors have addressed all the points raised by reviewers. No further comments.

Author Rebuttal, first revision:

NSMB-A47433A

We thank all reviewers and the editor again for their highly appreciative comments on our revised manuscript.

Reviewer #1:

None

Reviewer #2:

The authors have addressed all points raised at the highest possible standards.

Reviewer #3:

Nice revision. The authors have addressed all the points raised by reviewers. No further comments.

Final Decision Letter:

Message 25th Sep 2023

:

Dear Dr. Conrad,

We are now happy to accept your revised paper "Integrated chemical and genetic screens unveil FSP1 mechanisms of ferroptosis regulation" for publication as a Article in Nature Structural & Molecular Biology.

Your paper will be published online soon after we receive proof corrections and will appear in print in the next available issue. You can find out your date of online publication by contacting the production team shortly after sending your proof corrections. Content is published online weekly on Mondays and Thursdays, and the embargo is set at 16:00 London time (GMT)/11:00 am US Eastern time (EST) on the day of publication. Now is the time to inform your Public Relations or Press Office about your paper, as they might be interested in promoting its publication. This will allow them time to prepare an accurate and satisfactory press release. Include your manuscript tracking number (NSMB-A47433B) and our journal name, which they will need when they contact our press office.

About one week before your paper is published online, we shall be distributing a press release to news organizations worldwide, which may very well include details of your work. We are happy for your institution or funding agency to prepare its own press release, but it must mention the embargo date and Nature Structural & Molecular Biology. If you or your Press Office have any enquiries in the meantime, please contact press@nature.com.

If you have not already done so, we strongly recommend that you upload the step-by-step protocols used in this manuscript to the Protocol Exchange. Protocol Exchange is an open online resource that allows researchers to share their detailed experimental know-how. All uploaded protocols are made freely available, assigned DOIs for ease of citation and fully searchable through nature.com. Protocols can be linked to any publications in which they are used and will be linked to from your article. You can also establish a dedicated page to collect all your lab Protocols. By uploading your Protocols to Protocol Exchange, you are enabling researchers to more readily reproduce or adapt the methodology you use, as well as increasing the visibility of your protocols and papers. Upload your Protocols at

www.nature.com/protocolexchange/. Further information can be found at www.nature.com/protocolexchange/about.

Please note that *Nature Structural & Molecular Biology* is a Transformative Journal (TJ). Authors may publish their research with us through the traditional subscription access route or make their paper immediately open access through payment of an article-processing charge (APC). Authors will not be required to make a final decision about access to their article until it has been accepted. <https://www.springernature.com/gp/open-research/transformative-journals> Find out more about Transformative Journals

Sincerely,

Carolina Perdigoto, PhD
Chief Editor
Nature Structural & Molecular Biology
orcid.org/0000-0002-5783-7106

Click here if you would like to recommend Nature Structural & Molecular Biology to your

librarian:

<http://www.nature.com/subscriptions/recommend.html#forms>